# North Atlantic ventilation change over the past three decades is potentially driven by climate change

Haichao Guo [1,2] ✉, Wolfgang Koeve [1], Iris Kriest[1], Ivy Frenger [1], Toste Tanhua [1], Peter Brandt [1,3], Yanchun He [4], Tianfei Xue [1] & Andreas Oschlies [1,3]

The North Atlantic Meridional Overturning Circulation (AMOC) ventilates a large part of the world ocean via the formation of mode waters and North Atlantic Deep Water. The extent to which human activities have impacted this ventilation system remains unclear. To assess the temporal variations of ocean ventilation in the North Atlantic, we calculated the "age" of seawater, that is, the duration since its last contact with the ocean surface, from both observed and simulated chlorofluorocarbon-12 and sulfur hexafluoride concentrations. Our results indicate that, despite fluctuations in ventilation strength in the Labrador Sea over the past decades, the North Atlantic waters are generally aging. By integrating observations with model simulations, we propose that this aging trend is indicative of a climate change signal rather than natural variability.

The meridional overturning circulation in the North Atlantic is of great importance to the climate system due to its implications for the sequestration of excess heat and anthropogenic carbon and supplying dissolved oxygen to the ocean interior[1–5]. Its stoppage can result in strong and rapid climate shifts and has been labeled as one of the potential marine tipping points[6–8]. Considering the significant ice melting in the Arctic and Greenland over the past few decades[9], it is urgent to understand the temporal variations of the ventilation system in the North Atlantic under the changing climate[10].

Previous studies have extensively explored variations of ocean ventilation in the North Atlantic using various metrics, including transport velocity measured in the Rapid Climate Change program (RAPID,[11]) and Overturning in the Subpolar North Atlantic Program (OSNAP,[12]), sea surface temperature (SST) anomalies in the subpolar gyre relative to northern hemispheric or global SST[13,14], changes in dissolved oxygen concentration (e.g.,[15,16]) and convection depth (e.g.,[17–20]). However, due to the large inter-annual, decadal, and multi-decadal variability and relatively short period of observations[21,22], and also the uncertainty of indirect proxies in estimating AMOC (e.g., SST anomalies,[23,24]), it remains

uncertain if and how human activities have altered this ocean ventilation system.

Here, we use estimates of "water age" to quantify the temporal change of the ventilation process in the North Atlantic over the past three decades. Water age describes the time elapsed since the water was last in contact with the ocean surface. This property cannot be measured directly in the real ocean, but it can be estimated quantitatively from measurements of transient abiotic tracers such as chlorofluorocarbon-12 (CFC-12) and sulfur hexafluoride ($SF_6$) applying the Inverse Gaussian Transit Time Distribution (IG-TTD) approach (as detailed in "Methods"). IG-TTD assumes a distribution of the transit times of water masses from the ocean surface to the interior considering a combination of advective and diffusive transports[25,26]. In the following, "water age" refers to the mean age derived from the IG-TTD.

Compared to previous metrics used to assess changes of the AMOC, water age provides a unique advantage in studying changes in ocean ventilation. Unlike the transport velocity data from RAPID and OSNAP, transient tracer measurements offer broader temporal coverage dating back to 1981 and extend from tropical to high-latitude

[1]GEOMAR Helmholtz Centre for Ocean Research Kiel, Kiel, Germany. [2]Department of Oceanography, School of Ocean and Earth Science and Technology, University of Hawaii at Mānoa, Honolulu, USA. [3]Kiel University, Kiel, Germany. [4]Nansen Environmental and Remote Sensing Center, Bjerknes Centre for Climate Research, Bergen, Norway. ✉e-mail: hguo@geomar.de

regions in the North Atlantic (Fig. 1). In contrast to dissolved oxygen, which is also sensitive to biological activities, abiotic gases such as CFC-12 and $SF_6$ are influenced solely by ocean ventilation, which makes them particularly useful for tracing these processes. Previous studies have already utilized water age to investigate regional changes in ocean ventilation in the Southern Ocean[27,28], Nordic Sea[29], and the Arctic[30].

Building on prior work, our study introduces several methodological advancements. (i) We incorporate Δ/Γ constraints based on paired CFC-12 and $SF_6$ measurements during the IG-TTD calculations to improve robustness in identifying temporal changes[31]. Δ and Γ are the mean and width of the age spectrum, respectively[26]. (ii) Instead of focusing on specific sections, we grouped all observational data into three periods associated with different atmospheric variability modes[20]: the 1990s (1985–1999; positive North Atlantic Oscillation, NAO), 2000s (2000–2014; negative NAO), and 2010s (2015–2021; positive NAO). (iii) We also calculate the temporal change in Apparent Oxygen Utilization (AOU)-the difference between saturated and measured oxygen-as an additional metric to investigate ventilation variability. While AOU reflects accumulated respiration and can be influenced by biological activity[32], it remains a valuable indicator for understanding changes in ventilation. (iv) We subsample CFC-12 and

$SF_6$ data from seven Earth System Models and analyze the temporal evolution of ocean ventilation in the North Atlantic in models. This integrative approach provides valuable insights into the robustness of our observational interpretations on ventilation change. Details of our methodology are provided in the Methods section.

Despite the methodological advancements and the advantages of using CFC-12, it is important to note that, due to the relatively short atmospheric history of CFC-12, this tracer cannot effectively constrain the age of very old waters. We therefore focus on waters younger than 200 years, consistent with previous studies[33]. To assess ventilation changes in older water masses, additional measurements and simulations involving other tracers, such as Argon-39 and radiocarbon, will be required. However, these tracers are currently either not measured with sufficient spatial and temporal coverage[34] or not yet widely implemented in model simulations.

## Results

### Observed ventilation change in the North Atlantic over the past decades

Over the past three decades, the North Atlantic has seen substantial changes in ocean ventilation (Fig. 2, Table 1). We here present overall statistics from all measurements, without restricting the data to repeat

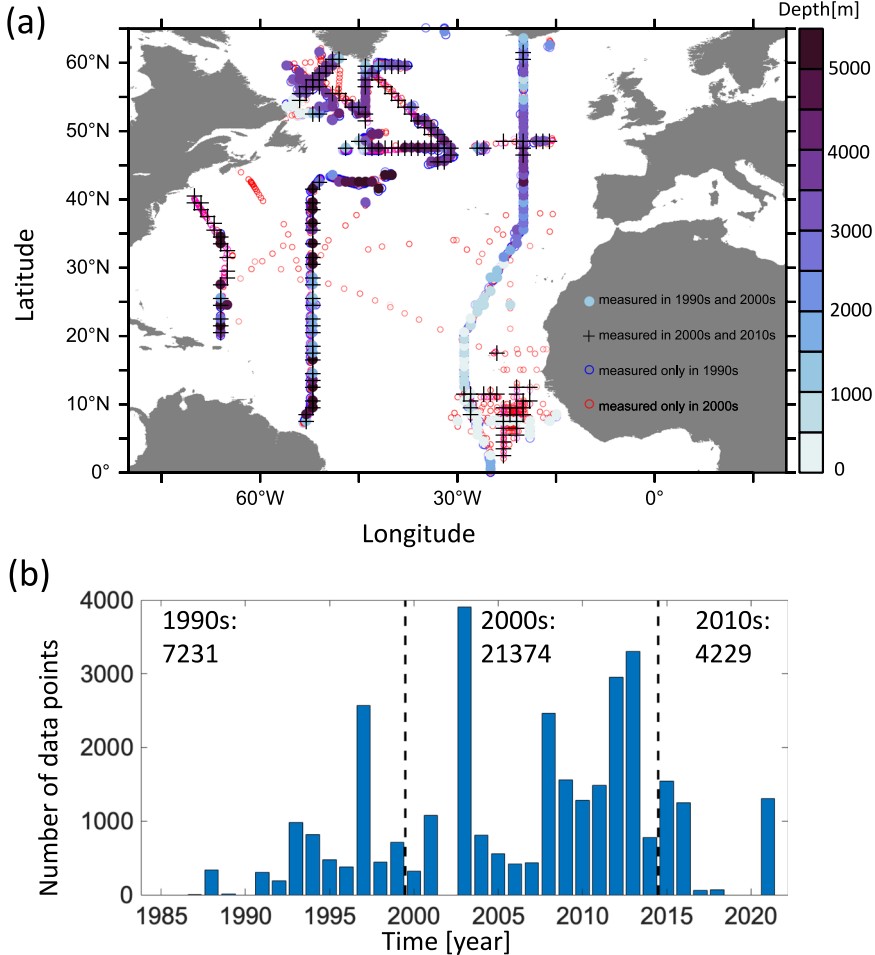

**Fig. 1 | Spatial distribution and temporal coverage of water age measurements in the North Atlantic. a** Spatial distribution of water age profiles from 1985 to 2021 in the North Atlantic. The filled circles present the profiles measured in both 1985–1999 and 2000–2014 periods, and the shading color in the filled circles indicates the maximum depth of each selected age profile. The cross symbol indicates the age profiles in both periods of 2000–2014 and 2015–2021. The open blue (almost none) and red circles indicate water age profiles measured only in

periods of 1985–1999 and 2000–2014, respectively. Measurements in the Caribbean Sea and the Mediterranean Sea have been excluded. **b** Annual count of water age measurements from 1985 to 2021 in GLODAPv2.2022. Dashed lines delineate the three time periods: 1985–1999, 2000–2014, and 2015–2021. The numbers of measurements suitable for age calculation in each period are 7231, 21374, and 4229, respectively (see "Method" for details).

# Observed change of ventilation

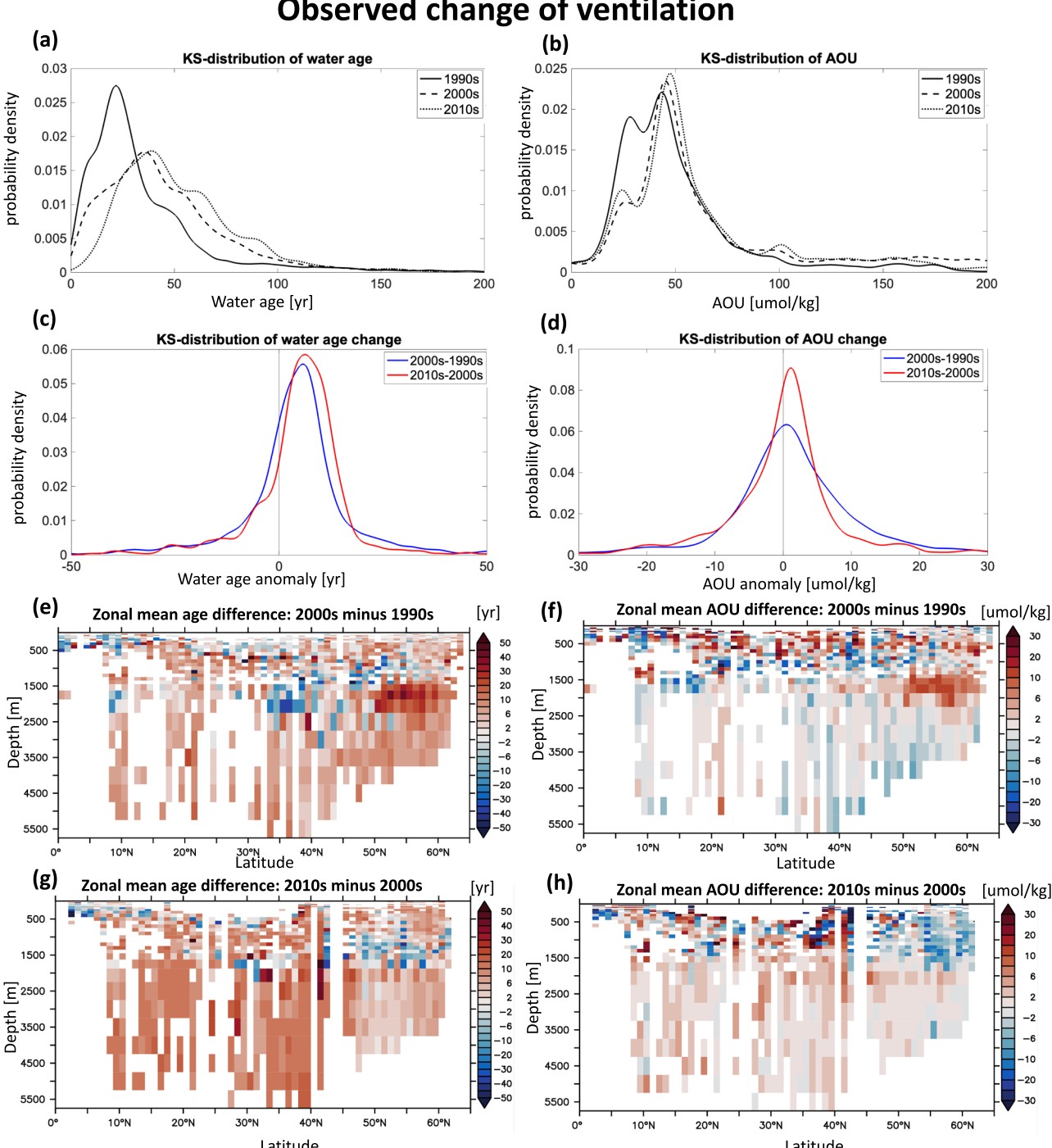

**Fig. 2 | Observed temporal changes in water age and apparent oxygen utilization (AOU) in the North Atlantic from the 1990s to the 2010s. a, b** Kernel Smoothing density estimates of water age (years) and AOU (μmol/kg) measured in the 1990s, 2000s, and 2010s for all sections. **c, d** Kernel Smoothing density estimates of the changes in water age (Δage) and AOU (ΔAOU) in the North Atlantic for repeated sections (blue: 2000s minus 1990s, red: 2010s minus 2000s). Zonal mean of Δage and ΔAOU in the North Atlantic, comparing the 2000s to the 1990s (**e, f**), and the 2010s to the 2000s (**g, h**). The data is from GLODAPv2.2022[59].

stations (Fig. 2a, b). A two-sample Kolmogorov-Smirnov test (by using *kstest2* function in MATLAB;[35,36]) at the 95% confidence level indicates significant differences in water age and Apparent Oxygen Utilization (AOU) among the 1990s, 2000s, and 2010s. Specifically, water age (volume-weighted mean) from the 1990s to the 2000s and from the 2000s to the 2010s increased by 6.8 years and 10 years, respectively. Other statistical metrics, such as non-weighted mean, median, and mode, show the same trend with varying magnitudes. Volume-weighted mean AOU also rose across these periods (Table 2).

Notably, in addition to temporal changes in ocean ventilation, the age differences among decades are also influenced by variations in sampling locations (Fig. 1) and the associated ventilation states.

Temporal changes in water age (Δage) and AOU (ΔAOU) at repeat-measure stations suggest an overall slowdown in ventilation from the 1990s to the 2010s in the North Atlantic (Fig. 2c, d). Kernel density estimates of Δage and ΔAOU indicate positive non-volume-weighted means, medians, and modes across periods (Tab. 1). The volume-weighted mean water age and AOU increased by 5.5 years and

0.8 µmol/kg from the 1990s to the 2000s, and by 6.6 years and 1.1 µmol/kg from the 2000s to the 2010s.

Despite the overall positive Δage in the North Atlantic, the Labrador Sea (48°N-60°N, 80°W-30°W) exhibits substantial variability in ventilation over different periods (Fig. 2e-h). From the 1990s to the 2000s, water age and AOU in this region decreased by −1.2 years at depths of 800–1200 m but increased significantly by +18.8 years at 1200 to 2500 m, likely due to a reduction in deep convection events, which previously extended to depths of around 2500 m in the early 1990s[37]. In the subsequent decade, from the 2000s to the 2010s, the shift from negative to positive NAO phases resulted in enhanced convective activity and water age decreases by −2.3 years in the 1000–1500 m range (Fig. 2g, h), consistent with previous findings[20]. This underscores the critical role of natural decadal variability in the ventilation process of the Labrador Sea, where intermittent deep convection events can obscure trends driven by external forces over shorter timescales. Further measurements of abiotic transient tracers over extended periods are crucial for gaining a better understanding of potential long-term anthropogenic impacts on ventilation in this region.

Besides the Labrador Sea, where natural variability is high, a consistent trend of aging was observed from the 1990s through the 2000s and into the 2010s in the rest of the North Atlantic (Fig. 2e, g). Interestingly, from the 1990s to the 2000s, some bottom waters in the North Atlantic experienced a decrease in AOU (Fig. 2e, g). We noticed that the co-occurrence of an increase in water age and a decrease in AOU has also been reported in previous studies in parts of the Nordic Sea (see[29] Fig. 8). We propose two possible explanations for the mismatch between increasing water age and decreasing AOU in the deep ocean. First, changes in ocean circulation within a warming climate may alter water mass composition, as different water types with varying biogeochemical histories are recombined over time, potentially introducing older yet less oxygen-depleted waters into a region[38]. Second, local biological activity can influence the AOU signal independently of ventilation: even if ventilation slightly decreases, reduced respiration rates, potentially driven by changes of primary production and export flux attenuation[39,40], could cause AOU to decline, reflecting changes of biological consumption rather than physical mixing changes. Further investigation is needed to disentangle these impacts. From the 2000s to the 2010s, both water age and AOU increased.

We estimate uncertainties in water age changes by considering both measurement and methodological uncertainties related to saturation assumptions. For measurement uncertainties, we performed 100 independent Monte Carlo simulations, adding random perturbations to CFC-12 measurements within a range of -3% to 3%, aligning with measurement precision[41]. Our analysis (Fig. S1) shows that, accounting for these uncertainties, the non-volume-weighted mean change in water age in the North Atlantic is $3.9 \pm 0.03$ years (mean $\pm$ standard deviation) from the 1990s to the 2000s, and $4.6 \pm 0.05$ years from the 2000s to the 2010s. This suggests that measurement uncertainties have a negligible effect on the detected temporal changes in water age.

For saturation uncertainties, we applied saturation levels from 90 to 50% in 10% increments during the IG-TTD calculations. This covers

**Table 1 | North Atlantic water age and Apparent Oxygen Utilization (AOU) and their temporal change across different periods**

| Variable | Periods | Mean | Median | Mode | Volume weighted mean |
|---|---|---|---|---|---|
| Water age (yr) | 1990s | 34.2 | 25.8 | 21.8 | 46.6 |
| | 2000s | 45.6 | 39.3 | 35.7 | 53.4 |
| | 2010s | 52.7 | 45.8 | 39.0 | 63.4 |
| AOU (µmol/kg) | 1990 | 50.1 | 43.4 | 43.5 | 60.8 |
| | 2000s | 73.0 | 50.9 | 45.2 | 67.3 |
| | 2010s | 65.0 | 49.7 | 47.5 | 68.2 |
| ΔWater age[a] (yr) | 2000s-1990s | +3.9 | +4.7 | +5.8 | +5.5 |
| | 2010s-2000s | +4.6 | +6.0 | +6.3 | +6.6 |
| ΔAOU[a] (µmol/kg) | 2000s-1990s | +1.8 | +0.9 | +0.5 | +0.8 |
| | 2010s-2000s | +0.3 | +0.7 | +1.1 | +1.1 |

All measurements (see Fig. 1) are used in the analysis.
[a]Only in regions where the age and AOU are measured in both periods.

**Table 2 | Temporal change (2000s minus 1990s) of North Atlantic water age and Apparent Oxygen Utilization (AOU) across multi-model mean and seven individual Earth System models**

| Variable | Models | Mean | Median | Mode | Volume weighted mean |
|---|---|---|---|---|---|
| ΔWater age (yr) | Multi-model mean[a] | +2.3 ± 1.2 | +2.3 ± 0.7 | +1.7 ± 1.1 | +2.3 ± 0.9 |
| | CanESM5 | +3.6 | +2.6 | +1.7 | +3.8 |
| | EC-Earth3-CC | +4.0 | +3.4 | +2.6 | +2.6 |
| | MRI-ESM2-0 | +2.6 | +2.5 | +1.4 | +2.2 |
| | NorESM2-LM | +2.1 | +2.5 | +2.6 | +2.7 |
| | NorESM2-MM | +1.3 | +2.6 | +2.8 | +2.4 |
| | UKESM1-0-LL | +0.7 | +1.2 | -0.5 | +1.1 |
| | FOCI-MOPS | +1.7 | +1.6 | +1.6 | +1.3 |
| ΔAOU (µmol/kg) | Multi-model mean[a] | +0.8 ± 0.3 | +0.7 ± 0.4 | +0.5 ± 0.3 | +0.6 ± 0.2 |
| | CanESM5 | +0.5 | +0.5 | +0.6 | +0.5 |
| | EC-Earth3-CC | +1.3 | +1.3 | +0.8 | +0.4 |
| | MRI-ESM2-0 | +0.4 | +0.3 | +0.2 | +0.7 |
| | NorESM2-LM | +0.8 | +0.8 | +0.9 | +0.8 |
| | NorESM2-MM | +0.9 | +0.5 | +0.4 | +0.6 |
| | UKESM1-0-LL | +1.1 | +1.1 | +0.2 | +1.0 |
| | FOCI-MOPS | +0.5 | +0.2 | +0.3 | +0.4 |

Model outputs are subsampled to match the spatial and temporal locations of observational data.
[a]For multi-models mean: mean ± one standard deviation.

# Historical change of ventilation in models

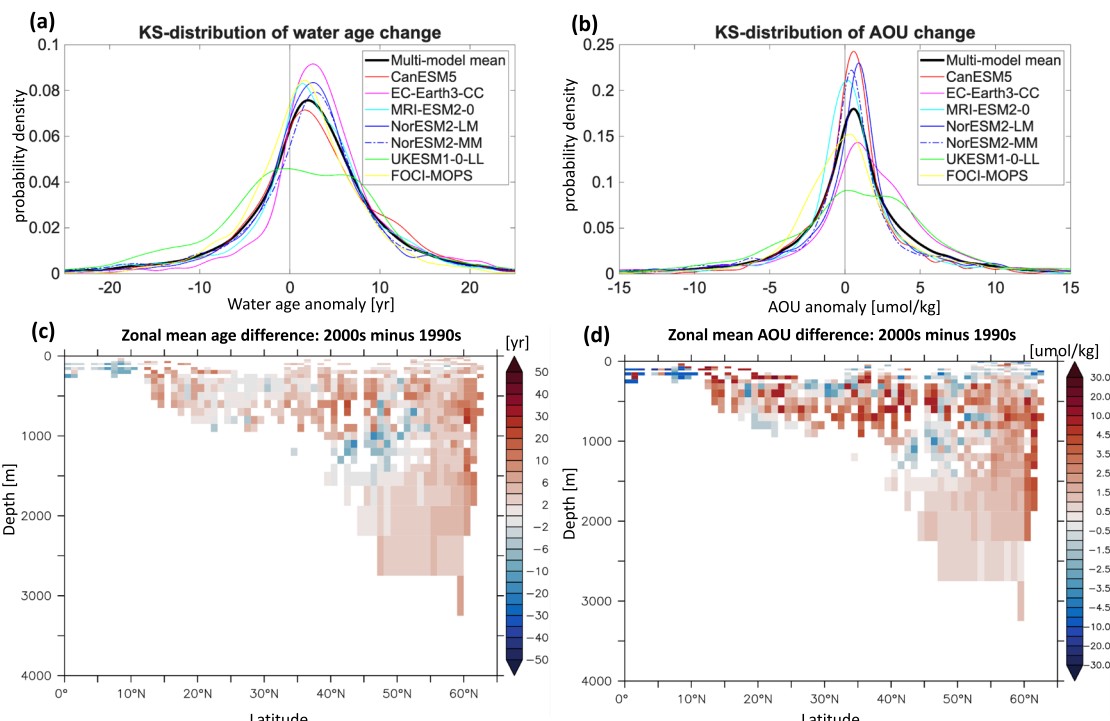

**Fig. 3 | Simulated temporal changes in water age and apparent oxygen utilization (AOU) in the North Atlantic from the 1990s to the 2000s. a, b** Display the Kernel Smoothing density estimation for the changes in water age (Δage, years) and AOU (ΔAOU, μmol/kg) in the North Atlantic, comparing data from the 1990s to the 2000s across seven individual models and the model average. Panels **c, d** present the multi-model mean of the zonal average for Δage and ΔAOU in the North Atlantic. The area covered by the multi-model mean (panels **c, d**) is smaller than that shown by the observational estimates (Fig. 2e, f), due to the slower spreading of abiotic transient tracers in models in the deep North Atlantic.

the range of saturation levels reported in previous observational studies[42–44]. We found that using a low saturation level, such as 50%, makes water age calculations for depths shallower than 1500 m in the Labrador Sea infeasible, as the partial pressure surpasses the highest atmospheric levels (see Fig. S2j). This indicates that such low saturation assumptions are impractical for these waters. Furthermore, despite the high sensitivity of Δage to saturation assumptions in Labrador Sea Water, the lower North Atlantic Deep Water consistently showed aging across all scenarios from the 1990s to the 2010s.

## Simulated ventilation change in the North Atlantic over the past decades

We subsampled the monthly (yearly for EC-Earth3-CC) average model outputs to match the spatial and temporal locations of observational data (see Methods for details). In the subsampled climate model simulations, water ages inferred from CFC-12 and SF$_6$ and AOU also increased in the North Atlantic covered by observations (Fig. 3). In the multi-model mean, the age of seawater and AOU have increased by +2.3 ± 0.9 years and +0.6 ± 0.2 μmol/kg from the 1990s to the 2000s, indicating the overall slow-down of ventilation under historical forcings. Kernel density estimates of Δage and ΔAOU for individual models show similar shapes and reveal a general trend of aging and oxygen depletion in the North Atlantic (Fig. 3a, b). Specifically, the age of water has increased by 1.1 years (UKESM1-0-LL) to 3.8 years (CanESM5) from the 1990s to 2000s, and AOU increased by 0.4 μmol/kg (EC-Earth3-CC and FOCI-MOPS) to 0.8 μmol/kg (NorESM2-LM) (Tab. 2). Despite this overall trend of aging, all individual models consistently underestimate the observed age increase, which is around 5.5 years. A detailed comparison between observed and simulated age anomalies suggests that the models fail to capture the pronounced age and AOU increases below 1500 m in the Labrador Sea (48°N-60°N,

80°W-30°W) from the 1990s to the 2000s (Fig. S3), likely due to their inability to simulate the intense deep convection of the early 1990s and its subsequent weakening. Further investigation is needed to confirm these discrepancies and understand their causes.

The zonally averaged Δage and ΔAOU generally show positive values in most ocean regions for the multi-model mean, indicating the general aging of waters in the North Atlantic in models (Fig. 3c, d). However, consistent with observations, certain areas—ranging from approximately 40°N to 50°N and between 800 and 1300 m depth—exhibit stronger ventilation, indicated by reduced water age by –3.0 years and AOU by –0.35 μmol/kg. Caution is advised when interpreting these patterns, as individual models often display anomalies of younger water age at various depths (Fig. S4). The distribution of counts for younger and older anomalies across depths and isopycnals further confirms that individual models show different patterns in age differences (Fig. S5). Nonetheless, despite younger age anomalies in parts of the North Atlantic, a larger volume of waters exhibits signs of aging from the 1990s to the 2000s across most depths and isopycnals, according to observations and all individual models.

Since the observations and corresponding subsampled model simulations are not temporally continuous, the temporal change of water age and AOU is also affected by natural short-term internal ocean tracer variability. To assess the anomalies induced by high-frequency variability, we conduct a comparison between the subsampled model outputs and the temporally full model outputs. Here, "temporally full" means we did not subsample the models' outputs according to the temporal coverage of observations. Results show that in each individual model used in this paper, the temporally subsampled model outputs effectively capture the variation in seawater age and AOU in the temporally full model outputs (Fig. S6, Table S2). In other words, the temporal coverage of observation in this

well-sampled region is sufficient to detect robust long-term variability or trends.

## Indications of anthropogenic effects on ventilation

Previous studies indicate that large interannual and decadal oscillations in the North Atlantic complicate efforts to understand how anthropogenic factors - such as sea ice loss, changes in heat flux, and wind patterns—affect the overturning circulation and ventilation[21,22]. Oscillations like the NAO have been shown to influence ventilation variability, particularly in the Labrador Sea[20,45]. However, ventilation can also be impacted by human-induced trends in surface ocean conditions, such as warming and freshening[46,47]. Based on abiotic transient tracer measurements and model simulations, we propose that ocean ventilation in the North Atlantic has been slowing since the 1990s through the 2000s and into the 2010s. The details supporting this conclusion are discussed below.

We investigate the evolution of water age in the North Atlantic across three periods characterized by shifts in the NAO phase, transitioning from positive (1990s) to negative (2000s) and back to positive (2010s)[20]. During both transitions - from positive to negative NAO and vice versa—we observe significant variability in ventilation within the Labrador Sea, consistent with previous findings[20,37]. However, despite the variability of water age in certain regions of the North Atlantic, the basin-scale mean water age and AOU continue to increase from the 1990s through the 2010s. This indicates that the North Atlantic's aging trend persists despite natural variability and phase shifts.

The decline in deep water ventilation observed from the 1990s to the 2000s appears to be a global phenomenon, extending beyond the North Atlantic. Independent studies have also reported increases in water age within the deep waters of the Southern Ocean[27] and the Nordic Sea[29]. This broader perspective enhances confidence that the changes in North Atlantic ventilation represent a long-term trend driven by climate change. While previous studies have also reported enhanced ventilation in certain mode or intermediate waters, such as Subantarctic Mode Water[27] and Arctic Intermediate Water[29], our findings highlight notable regional differences-for example, waters became younger in the upper eastern basin and older in the upper western basin of the North Atlantic. Measurements of abiotic transient tracers over larger spatial and temporal scales are required for a better understanding of the change of ventilation in mode waters in the North Atlantic.

Furthermore, results from historical model simulations support the interpretation that observed ventilation changes constitute a climate change signal. Despite varying magnitudes, all models indicate a general aging trend of North Atlantic water (Table 2). Considering the differing phases of the NAO across models (Fig. S7), the robustness of this trend underscores that it is a significant climate change indicator. Notably, we used seven Earth System Models in this study, as to our knowledge, these represent all available CMIP6 simulations that provide the necessary variables for our analysis. We acknowledge that the limited number of models may not fully capture internal variability and could introduce uncertainty in the estimated ventilation trends. Future studies employing larger model ensembles will help further constrain this uncertainty.

To further investigate how ocean ventilation responds to global warming, we analyze the projections of water age in models under the high carbon dioxide emission scenario (the Shared Socioeconomic Pathways 5 8.5 scenario, SSP5-8.5,[48]) where the climate change signals are amplified. By the end of the century, all models predict increased ventilation in the upper low latitudes (0°N-15°N, 0-500 m), with a notably stronger signal in the NorESM2-LM and NorESM2-MM models (Fig. 4, Fig. S8). This pattern of enhanced ventilation in upper low-latitude waters has been reported previously[49,50], with the explanation that such enhancement results from reduced mixing with upwelled old waters under warming conditions. Moreover,[51] propose another explanation that the thinning of isopycnic layers in the upper ocean during a warmer climate may facilitate enhanced ventilation rates of central waters (see their Fig. 2 for details). Despite these regional variations, an overall pattern of ocean aging emerges in the North Atlantic, particularly in the deep ocean and high latitudes (Fig. 4). Further analysis under the SSP5-8.5 scenario confirms that this pattern persists across different time periods (e.g., 2090s-2010s and 2090s-2020s), indicating that it is unlikely to be driven by internal variability. In the end, the models suggest a general slowdown of ocean ventilation in the North Atlantic associated with warming as a climate change signal.

## Discussion

Using measurements of abiotic transient tracers from 1985 to 2021, we identified temporal changes in ocean ventilation in the North Atlantic. Our results suggest that, despite variability in ventilation strength within the Labrador Sea, the water in the North Atlantic—particularly in deeper and high-latitude regions—gradually ages. By integrating observations across broader ocean basins and model simulations, we propose that this persistent aging is a climate change signal, with indications of: (i) an ongoing increase in water age and AOU during both phases of NAO transition— from positive to negative and vice versa; (ii) reports of deep-water aging in other ocean basins; (iii) consistent evidence across seven Earth System models, which, despite differing patterns of ventilation change, collectively support a trend of North Atlantic water aging; and (iv) an intensification of aging of North Atlantic water under future climate projections under high-emission scenarios.

Such a slow of ocean ventilation in the North Atlantic is likely to have significant impacts on the climate system, especially concerning the carbon and oxygen cycles. Many studies have shown that ocean ventilation plays a major role in regulating the global carbon cycle: in part directly, via anthropogenic carbon storage (e.g.,[3,52–54]), and in part indirectly, via storage of dissolved inorganic carbon associated with the biological carbon pump[54,55]. The slowdown of ocean ventilation is supposed to increase storage of carbon and nutrients by the biological pump[55,56] and decrease the anthropogenic carbon uptake by the solubility pump[56], with net effects of reduction of marine carbon sink. For oxygen, weaker ventilation, without considering the biogeochemical feedback in the water column, results in ocean deoxygenation and may threaten marine ecosystems[54,57]. According to current models, the reduced ocean ventilation in the deep ocean is committed to continue for several hundred years, even if anthropogenic carbon dioxide emissions are halted immediately or if atmospheric carbon dioxide levels recovered to the pre-industrial level[54,57,58]. As a consequence marine ecosystems may face long-term ocean deoxygenation.

## Methods
### Observations

We used the Global Ocean Data Analysis Project products [[59]GLODAPv2.2022: https://www.ncei.noaa.gov/access/ocean-carbon-acidification-data-system/oceans/GLODAPv2_2022/] which provides potential temperature, salinity, concentrations of anthropogenic transient tracers (including CFC-12 and $SF_6$ in unit of pmol/kg and fmol/kg, respectively) and also other biogeochemical tracers (e.g., $O_2$ in unit of µmol/kg). This data product is composed of data from 1085 scientific cruises covering the global ocean, and contains more than 300,000 quality-controlled measurements of CFC-12 and 66,000 valid measurements of $SF_6$ during the period between 1981 and 2021, with most of the $SF_6$ measurements taken after 2000. We select measurements of CFC-12 concentration and $SF_6$ concentration that are above their detection limit of 0.01 pmol/kg and 0.1 fmol/kg, respectively[60]. The potential temperature, salinity, and $O_2$ in this dataset is resampled

# Change of ventilation by the end of the century

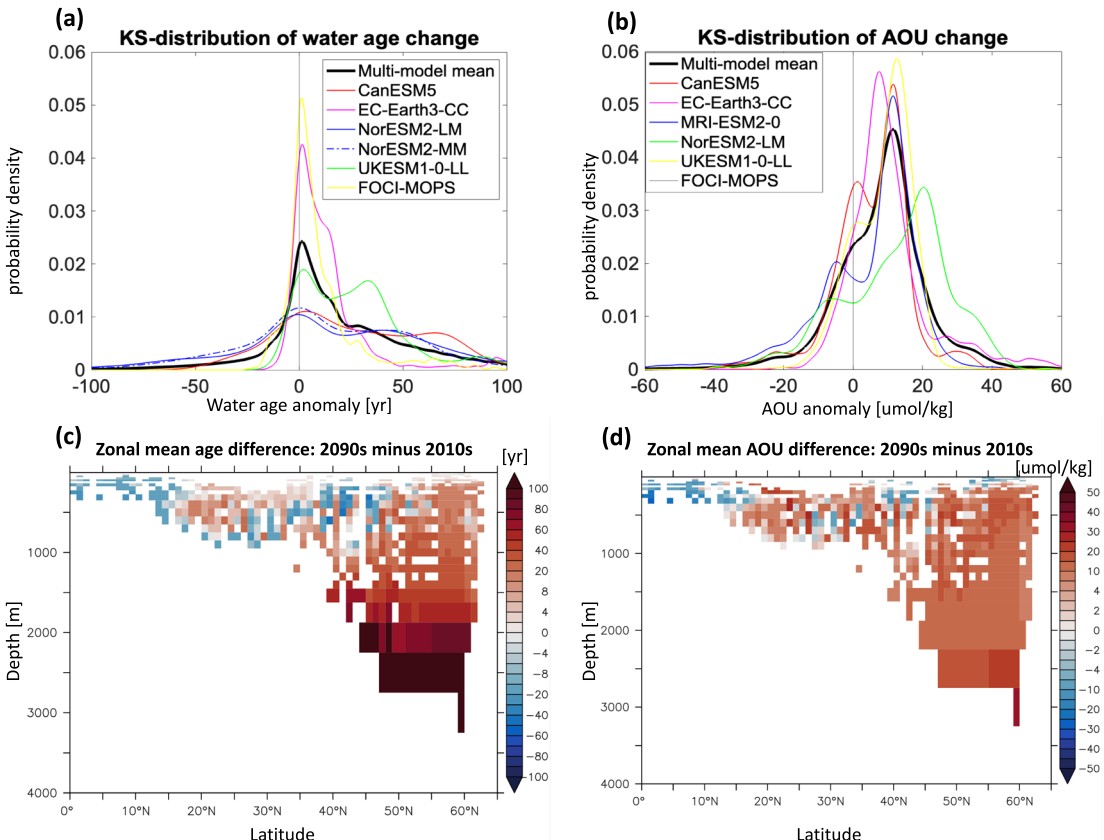

**Fig. 4 | Simulated temporal changes in water age and apparent oxygen utilization (AOU) in the North Atlantic from the 2010s to the 2090s. a, b** Display the Kernel Smoothing density estimates of the changes in water age (Δage, years) and AOU (ΔAOU, µmol/kg) in the North Atlantic, comparing data from the 2010s to the 2090s across individual models and the multi-model mean. Panels **c, d** show the multi-model mean of the zonal average for Δage and ΔAOU in the same region. In these analyses, the ideal age is used to detect ventilation changes; it functions like a "clock," resetting to zero at the sea surface and increasing by one day per day in the sub-surface layers. Note that ideal age data in MRI-ESM-2-0 and dissolved oxygen concentration data in NorESM2-MM were not accessible for the SSP5-8.5 scenario during the course of our study (accessed in August of 2025, https://esgf-metagrid.cloud.dkrz.de/search/cmip6-dkrz/).

analogously to the observational CFC-12 and $SF_6$ in the GLO-DAPv2.2022 dataset.

## Dual-tracer constrained inverse Gaussian transit time distribution

Water age in the main text refers to the mean age of Inverse Gaussian Transit Time Distribution (IG-TTD) inferred from measurements of transient tracers. TTD is a distribution of the transit times of water masses from the ocean surface to the interior considering a combination of advective and diffusive transport pathways[25,26]. In brief, the concentration of any transient passive tracers (e.g., CFC-12), $c(\mathbf{r}, t)$, at location $\mathbf{r}$ and time $t$ can be related to its surface history and its transit-time distribution, such as:

$$c(\mathbf{r}, t) = \int_0^\infty c_0(t - \xi) G(\mathbf{r}, \xi) d\xi \qquad (1)$$

where $c_0(t - \xi)$ is the history of tracer concentration at the surface ocean $\xi$ years before the time of observation $t$. For CFC-12 and $SF_6$ used in this study, their atmospheric history has been observed from 1936 to 2015[61], which provides the boundary condition $c_0$. The TTD $G(\mathbf{r}, \xi)$ at location $\mathbf{r}$ can be estimated from CFC-12 measurements under the conditions of (a) TTD takes the shape of an Inverse Gaussian function (IG, expressed as equation (2)), (b) the ratio of the width ($\Delta$, the centered second moment of $G(\mathbf{r}, \xi)$ as expressed in equation (4)) to

mean age ($\Gamma$, the first moment of $G(\mathbf{r}, \xi)$ as expressed in equation (3)) of the IG distribution is specified[26], and (c) the saturation of transient tracer is 100%.

$$G(\mathbf{r}, \xi) = \sqrt{\frac{\Gamma^3}{4\pi\Delta^2\xi^3}} \exp\left(\frac{-\Gamma(\xi - \Gamma)^2}{4\Delta\xi}\right) \qquad (2)$$

$$\Gamma = \int_0^\infty \xi G(\mathbf{r}, \xi) d\xi \qquad (3)$$

$$\Delta = \frac{1}{2} \int_0^\infty (\xi - \Gamma)^2 G(\mathbf{r}, \xi) d\xi \qquad (4)$$

In our IG-TTD calculations, we applied $\Delta/\Gamma$ constrained by the combination of CFC-12 and $SF_6$ paired measurements, which help TTD provide a robust temporal change in water age[31]. In detail, firstly, we selected all samples where both CFC-12 and $SF_6$ were measured. Then, we calculated $\Gamma$ for CFC-12 and $SF_6$ individually, varying $\Delta/\Gamma$ from 0.2 to 1.8 at intervals of 0.1. The local optimal ratio is chosen as the one that minimized the difference in $\Gamma$ derived from CFC-12 and $SF_6$. We chose an upper limit of $\Delta/\Gamma$ of 1.8 because $\Delta/\Gamma$ exceeding 1.8 causes the water age inferred from IG-TTD to become highly sensitive to the assumption of tracer saturation levels at the time of water mass formation and cannot be well constrained by the CFC-12 and $SF_6$ pair[60]. The

**Table 3 | Earth system models used in this study, including data references; their individual ocean, sea-ice, and marine biogeochemistry components; and the assessed simulations**

| Model and reference | Ocean and sea ice | Marine biogeochemistry | Simulations |
|---|---|---|---|
| CanESM5[70,71] | NEMO3.4.1 + LIM2 | CMOC | historical + ssp585 |
| EC-Earth3-CC[72,73] | NEMO3.6 + LIM3 | PISCES | historical + ssp585 |
| MRI-ESM2-0[74,75] | MRICOM4 | NDZP | historical |
| NorESM2-MM[76,77] | BLOM + CICE5.1.2 | iHAMOCC | historical + ssp585 |
| NorESM2-LM[76,78] | BLOM + CICE5.1.2 | iHAMOCC | historical + ssp585 |
| UKESM1-0[79,80] | HadGEM3-GC3.1 | MEDUSA | historical + ssp585 |
| FOCI[65] | NEMO3.6 + LIM2 | MOPS | esm-hist + esm-ssp585 |

constrained $\Delta/\Gamma$ was then applied to the time series of CFC-12 measurements and subsampled model simulations from the time period 1985–2018 for IG-TTD calculation, assuming the temporal consistency of $\Delta/\Gamma$[31]. For more technical details and uncertainty analysis of the constrained IG-TTD approach, readers are referred to[60,62], and[31].

### Data binning
To analyze the temporal change of water age, we have binned the observational data into three periods with different atmospheric variability modes[20]: 1985–1999 (referred to as 1990s in the following), 2000–2014 (referred to as 2000s in the following), and 2015–2021 (referred to as 2010s in the following). Afterward, we have placed water age into bins which are the $1° \times 1°$ horizontal grid cells with 33 standard depths of the GLODAPv2.2016b[63]. Eventually, the binned water age profiles, which have been measured at least once both in the 1990s and 2000s, or both in the 2000s and 2010s, are selected to quantify anomalies of the water age (Fig. 1). We excluded the measurements in the Nordic Sea (north of Iceland) because a deliberate tracer release experiments and subsequent spreading of large amounts of $SF_6$[64] limit the application of $SF_6$ to constrain the IG-TTD. A recent study suggests that the ventilation in the Nordic Sea generally became stronger over the past decades except in the deep waters[29].

### Models subsampling
For the ocean models, we have taken the historical simulations from the Flexible Ocean and Climate Infrastructure (FOCI,[65,66]), and 6 ESMs that contributed to CMIP6 (Table 3) from https://esgf-data.dkrz.de/search/cmip6-dkrz/. These models simulate all key variables required for this study (potential temperature, salinity, $O_2$, CFC-12, $SF_6$, and idealized age tracer). We did not calculate the drift due to the (i) lack of pre-industrial simulations of CFC-12 and $SF_6$, (ii) difficulties in locating corresponding pre-industrial simulations that align with existing historical simulations.

To facilitate model-data comparison, as in previous studies (e.g.,[67]), model outputs were bilinearly interpolated to a horizontal $1° \times 1°$ grid and then linearly interpolated to the vertical axis of 33 standard depths of the GLODAPv2.2016b[63], via the climate data operator (CDO,[68]) *remapbil* and *intlevel* functions. Then we subsampled the monthly (yearly for EC-Earth3-CC) average output of model simulations analogously to the observational sampling schedules, i.e., at the same spatial and temporal locations of observations. To align the units of variables between CMIP6 model outputs and GLODAPv2.2022, we convert the units of CFC-12, SF6, and $O_2$ from $mol/m^3$ (unit used in CMIP6 simulations) to pmol/kg, fmol/kg, and µmol/kg by using a constant density of seawater, i.e., $1025 \, kg/m^3$[67]. The subsampled model outputs are processed following the same procedure as for observations. Notably, due to the inherent limitation of most of our simulation outputs being available only at a monthly temporal resolution and coarse spatial resolution (e.g., $1° \times 1°$), we were only able to subsample the model simulations to match the observation at such a relatively coarse scale, rather than achieving a finer temporal and spatial scale alignment.

## Data availability
All CMIP6 data used here are available from: https://esgf-node.llnl.gov/search/cmip6/. Observational data, the Global Ocean Data Analysis Project products GLODAPv2.2022, is from[59]: https://www.ncei.noaa.gov/access/ocean-carbon-acidification-data-system/oceans/GLODAPv2_2022/. The FOCI data and material that support the findings of this study are available through GEOMAR at https://hdl.handle.net/20.500.12085/8cc3d1a6-4df9-408e-971e-43a70f335282[69].

## Code availability
The FOCI model code is provided by[66] at https://doi.org/10.5281/zenodo.6772175. The scripts used for data processing and for producing the figures presented in this manuscript are available at: https://hdl.handle.net/20.500.12085/8cc3d1a6-4df9-408e-971e-43a70f335282[69].

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

## Acknowledgements

We gratefully acknowledge the insightful discussions with colleagues from the Biogeochemical Modeling research unit at GEOMAR. In particular, we thank Markus Schartau and Mathias Zeller (GEOMAR), Ilaria Stendardo (Bremen University), and Xuenan Li (Kiel University) for their valuable contributions to the data analysis. We also thank the three anonymous reviewers for their thorough and constructive comments, which significantly improved the quality of this paper. I.F acknowledges funding by the European Research Council (ERC) Grant (101116545; https://doi.org/10.3030/101116545). We acknowledge the use of the Ferret program developed by NOAA's Pacific Marine Environmental Laboratory for the analysis and graphics presented in this work. This study is a contribution to Subtopic 6.3, The future biological carbon pump, of the Helmholtz Research Program Changing Earth—Sustaining our Future.

## Author contributions

H.G. conceptualized the research, generated the figures, and wrote the initial manuscript under the supervision of A.O. W.K., I.K., I.F., T.T., P.B., Y.H., and T.X. contributed to the analysis and interpretation of the results. All authors revised the manuscript and approved the final version.

## Funding

## Competing interests

The authors declare no competing interests.
