## [Transparent Peer Review file · Nature Communications]

North Atlantic ventilation change over the past three decades is potentially driven by climate change

Corresponding Author: Dr Haichao Guo

Version 0:

Reviewer comments:

Reviewer #1

(Remarks to the Author)
Please see the attachment.

Reviewer #2

(Remarks to the Author)

Guo et al. present an analysis of historical transient tracer (CFC-12 and SF6) measurements from the North Atlantic to constrain an inverse model (Inverse Gaussian Transit Time Distribution; IG-TTD) to explore changes in apparent ventilation age since the early 1980s. Grouping the dataset (GLODAPv2) into three time periods, they find an overall increase in water age in the deep North Atlantic, and decrease in mean water age in the intermediate North Atlantic, suggesting a reduction in deep-ocean ventilation in the North Atlantic. The sign of deep- and intermediate-ocean changes in ventilation ages are consistent with simulations from several Earth System models, but in terms of ideal mean age (simulated directly from the models) and the age inferred from applying the IG-TTD approach to model-simulated CFC-12 and SF6. Overall the paper is well written and presented in a rather accessible format. The conclusions appear to be robust, but I would prefer to see (a) a more quantitative analysis of changes in inferred water mass age in the deep and intermediate North Atlantic, and (b) a more complete error analysis. I realize that much of the IG-TTD method and associated analysis is presented in a companion paper (Guo et al., 2024; preprint available via EGU sphere), which is focused on large-scale changes in water mass age throughout the global ocean and includes an analysis of, e.g., the sensitivity of results to the assumption of complete saturation at the time of air-sea exchange, or the choice of Δ/Γ . However, for this paper and its specific focus changes in ventilation age in the climatically important North Atlantic, I would like to see more sensitivity testing to fully convey the uncertainty on these main findings. The manuscript is timely, broadly relevant to global climate change, and well formulated. I would recommend publication in Nature Communications after the authors address these issues, which I describe in more detail below.

1. The main conclusions about changes in intermediate and deep North Atlantic ventilation age and other water mass properties (e.g., potential density, salinity, AOU) are described qualitatively and shown (quantitatively) in the figures (e.g., Figs. 2 and 3). I think it would strengthen the overall conclusions of the paper to provide average values (and uncertainties) of these properties and ages – and their temporal changes – for the water masses being discussed. For example, by defining intermediate and deep waters using either depth or potential density, the authors could estimate a mean change in ventilation age for age water mass, along with corresponding mean changes in salinity, potential density, potential temperature, and AOU. It would be helpful and provide more confidence in the ultimate result for the authors to be able to state that, e.g., the mean ventilation age of the deep North Atlantic increased by $x \pm y$ years from the 1990s to 2010s. Similarly, providing mean changes in these properties in the ESMs would be very helpful, while recognizing that the timescale of ESM simulations is different.

2. Relatedly, I would like to see a formal error analysis on the main result, accounting for the sensitivity of IG-TTD ages to key factors like the choice of initial saturation and Δ/Γ . Could a monte carlo simulation be performed to estimate the uncertainty in the changes of ventilation age (and pot. temp., pot. density, salinity, AOU) for mean water masses (e.g., deep and intermediate North Atlantic)? For example, carrying out many independent monte carlo simulations using a range of

assumptions about Δ/Γ and saturation, while also adding random perturbations to CFC-12 and SF6 measurements to account for measurement uncertainty. This would be an important addition to evaluate the robustness of the main result, which is presently hard to do based solely visual representation of the results in the figures.

3. Are ESM simulations available running from the pre-industrial era through 2100? Or just 2015 to 2100? If the former are available, this would be a more useful comparison with the observational data set (e.g., binning ESM results into the same three time periods from the 1980s to present)

Reviewer #3

(Remarks to the Author)
Please see the attachment.

Version 1:

Reviewer comments:

Reviewer #1

(Remarks to the Author)

In the revised manuscript, authors have sharpened the focus on water age and AOU and included discussions on uncertainties regarding temporal sampling and saturation levels, which enhances the clarity of the manuscript and the robustness of the findings. Even so, the conclusions could be strengthened by addressing the following key points:

1. The discussions on water age decadal variability and trend are better explained in the revised manuscript. While the decadal contrast between 1990s and 2000s is quite clear, that between 2000s and 2010s remains less convincing. The top reason is that data used for 2010s is from 2015-2021, covering only 7 years. This time period is too short to robustly represent the mean ventilation strength during the decade, given that the transit time for deep ventilated waters to lower latitudes exceeds 10 years (e.g. an average export time scale for overflow waters is ~20 years based on high-resolution ocean model; Lozier et al., 2013). To address this inherent limitation of the observational record, the authors could use model output to reconstruct a longer historical time series of deep water age or AOU to better contextualize the trend.

2. While the uncertainty from temporal sampling is now discussed, the potentially significant impact of spatial sampling biases—raised in the previous review—warrants similar investigation. Testing the results against model data subsampled at the observational spatial locations would greatly strengthen the analysis.

3. In observations, aging trend is dominated by waters below 2000m, where model output is mostly lacking. Instead, aging trend in models is concentrated in the upper 1000m. This model-observation difference needs to be elaborated because it directly relates to the type of water mass experiencing ventilation change (e.g. mode water or deep/abyssal water). Quantifying and comparing trends above and below 2000m separately would help clarify the dominant layers of ventilation change and reconcile this difference.

Minor comments

4. Line 68: Specify Δ/Γ in text.

5. Line 71: Specify what the modes are.

6. Figure 2: Please clarify in figure caption that (a-b) are for all sections while (c-d) are for repeat sections.

7. Please acknowledge the limited number of models used may introduce uncertainties to the trend signal.

Reference

Lozier, M. S., Gary, S. F., & Bower, A. S. (2013). Simulated pathways of the overflow waters in the North Atlantic: Subpolar to subtropical export. *Deep Sea Research Part II: Topical Studies in Oceanography*, 85, 147–153.

Reviewer #2

(Remarks to the Author)
The authors have satisfactorily responded to my initial comments.

Reviewer #3

(Remarks to the Author)

I appreciate the effort the authors have put into revising the manuscript and generally find my comments well addressed. The added, more quantitative analysis makes the manuscript much stronger.

That said, I share some of the concerns raised by Reviewer #1 in the first round regarding whether the presented results and analyzed data provide sufficient evidence to conclusively attribute the reported decadal ventilation changes in the North Atlantic to climate change. However, I deem the results interesting and worth publishing, as the authors have added critical

discussion on the robustness of their results and the inherent limitations of available observations and model results, and have consequently phrased their conclusions more carefully. Considering this, I wonder whether it would be more appropriate to also reflect this in the title by phrasing it as a question, i.e., “Variation of ventilation in the North Atlantic over the past three decades - a climate change signal?”

Below are a few additional comments that I would appreciate seeing addressed:

Line 68: This seems to be the first usage of the symbols delta and gamma. Please explain here what they mean.

Line 70: “... we used all observational data into ...” >>> something is wrong with this sentence

Lines 75-76: “Although we noted AOU as a measure of accumulated respiration that biological activities can influence (Buchanan & Tagliabue, 2021), it still contributes ...” >>> wording needs clarification

Line 85: Briefly explain or give reference to Kolmogorov-Smirnov test

Figure 2a: Do you have an explanation why the changes from 1990s to 2000s are much more significant than from 2000s to 2010s? I presume it has to do with sampling locations? You mention this in lines 91-93, but I believe it could be made clearer.

Lines 109-111: Sentence “This underscores...” is somewhat unclear. I can guess the meaning, but the wording is rather confusing (e.g., what acts on shorter time scales here, and why should deep convection not be influenced by external forcing?)

Line 144: “such as 50%” instead of “like 50%” ?

Lines 155-156: The expression “In the mean of models ocean, ...” sounds wrong

Lines 163-165: Could you provide some informed speculations about the potential reasons for these discrepancies in all models?

Lines 172-173: Concerning “The distribution of counts ...”: I am wondering if a vertically varying model resolution would in any way impact the shape of these distributions. Same for other distributions that are based on “counts”, e.g., Fig. 2a-d. Or is this irrelevant because of how you bin the data to standard depths and how the distributions are then constructed?

Figure 3a,b and similar KS-distributions: Consider adjusting the x axis range to make the relevant regions clearer as the pdfs are essentially 0 from +-25 yr and +-15 umol/kg outward. As it is, there is a lot of wasted white space in the figure and the relevant parts are unclear.

Figure 3d: y axis label and tick labels overlap

Caption of Figure 3: In second line, add space between “and” and “Δ”

Lines 186-187: “... temporal coverage of observations ...”

Lines 191-192: “sea ice loss” instead of “ice melting”? (as ice melting occurs naturally all the time)

Line 207 and other relevant passages throughout the manuscript: You generally refer to basin scale (as in full depth) mean water age. Is that really the best metric to assess ventilation changes? I would assume intermediate or deep water age might be more appropriate and might yield quite different values for the mean aging trends. Or is this irrelevant because of the way you calculate age? Could you please comment on this?

Line 204: Drop “For the real ocean,”

Line 307: “... in the main text ...”

Lines 352-353: You discuss the limitations of the method in general, but do not elaborate what this implies for your results. For example, does this sentence imply that your age estimates should be seen as lower bounds and that the actual age is likely larger in regions where the real distributions are multi-modal?

Table 3: In the third row, fix citation and reference (line 431). “Consortium” is not a last name but part of the full name of the consortium. If I am not mistaken, this is the EC-Earth Consortium (EC-Earth)

Version 2:

Reviewer comments:

Reviewer #1

(Remarks to the Author)

Authors have nicely and carefully addressed my concerns. I thus recommend immediate publication.

Reviewer #3

(Remarks to the Author)

The authors have addressed all my previous comments.

I do believe a question mark in the title would make it more accurate. However, as this is not supported by the journal, I will suggest the following alternatives but leave it up to the authors and the journal editors to work this out:

Variation of ventilation in the North Atlantic over the past three decades – ...

... Evidence for a climate change signal

... Indications of a climate change signal

... A potential climate change signal

Below are a few additional minor comments/suggestions/corrections:

Main text:

Figure 1: Depth colorbar labels are barely readable

Figure 2a-d: Panel labeling overlaps with y-axis tick labels

Most figures where y-axis shows depth: depth >>> Depth [m]

Figure 4d: y-axis label overlaps with tick labels

Minor aesthetic note: Panel labeling in figure captions is very inconsistent. E.g., Figure 2. (a, b); Figure 3. Panels (a and b);

Figure 4. Panels (a) and (b)

SI:

Figure S3: Unclear over what periods the differences are calculated.

Figure S4 caption: typo “acorss”

Figure S6: “The same as Fig. 3” >>> Should this be Figure S4 with the revised figure numbering?

Figure S7 caption: “The NAO index are” >>> mixed plural and singular

Figure S8 caption: typo “acorss”

The authors thank the reviewers for their very helpful comments and suggestions. We here provide a point-by-point response to all reviewers. The reviewers' comments are given in black, and our responses are given in blue. We refer to line numbers in our revised manuscript.

Reviewer #1 Comments to Author:

Summary

Using observed and simulated transient abiotic tracers, the study investigates changes of ventilation in the North Atlantic in the 1990s, 2000s and 2010s. They find enhanced ventilation of the intermediate waters and reduced ventilation of the deep waters since the 1990s. These decadal changes are further regarded as a climate change signal in response to external forcing.

The study focuses on an interesting topic, but I am not convinced that the decadal ventilation changes in the North Atlantic are a climate change signal. This is because: (1) observed hydrography clearly shows a reduction of ventilation of Labrador Sea intermediate waters in the 2000s, which is part of a significant decadal variability, contradicting the argument by the current study; and (2) climate model results are insufficient to separate internal variability from a climate change signal. As such, I cannot recommend publication. Below I list my major comments.

A: We appreciate your constructive feedback and acknowledge the critical role of decadal variability in the ventilation process, particularly in regions like the Labrador Sea where internal decadal to multidecadal variability as well as intermittent deep convection events can obscure trends associated with anthropogenically forced changes. In the revised manuscript, we have emphasized this point more clearly. Our results suggest that despite fluctuations in ventilation strength in the Labrador Sea over the past decades, most of the North Atlantic waters, particularly at depth, are consistently aging. By integrating observations with an ensemble of model simulations, we propose that this aging trend is indicative of a climate change signal rather than natural variability.

Additionally, in the original submission, our focus was spread across multiple hydrographic variables such as temperature and salinity, which may have diluted the emphasis on ventilation. In the updated manuscript, we have shifted our focus specifically on water age and AOU, providing clearer and more direct insights into ventilation changes. These revisions strengthen the manuscript and better support our key findings.

Major comments

(1). The reported enhanced ventilation in the intermediate layer of the Labrador Sea in the 2000s contrasts previous studies using hydrography. By comparing water age in the 2000s to the 1990s (Figure 3g), it is argued that age of intermediate water (1000- 2000 m) decreases, suggesting an enhanced ventilation in the Labrador Sea. However, hydrography clearly shows a warmer, saltier and lighter intermediate layer in the 2000s, which reflects weakened ventilation. This reduced ventilation in the 2000s is well documented in many observational studies (e.g. Yashayaev, 2024).

Actually, the age change between 1000 m and 2000 m in the Labrador Sea shown in Figure 3g seems to be positive to me. The color scale makes it difficult to read.

A: We appreciate the reviewer's attention to this error along the lines of 140 - 141. We have clarified in the revised manuscript that, in the Labrador Sea, water age decreases occur at depths of 800m to 1200m during the 1990s to 2000s and from 1000m to approximately 1500m during the 2000s to 2010s (lines 98-111). At lower latitudes, the presence of younger water is observed reaching depths of 2000m.

Furthermore, we agree and our results show this, that the water age at depths of 1200m to 2500m in the Labrador Sea has significantly increased from the 1990s to the 2000s (Figure 2e in the revised manuscript, also see below). This likely results from deep convection events occurring in the early 1990s but not thereafter. We have noted this in the revised manuscript (Lines 98-110), and it is consistent with findings by Yashayaev and Clarke (2008) and Yashayaev I (2024).

We apologize for the difficulty in reading the color scales in Figure 3 (Figure 2 in the revised manuscript, also see below), and we have now updated our figures to make them more readable. Moreover, in the revised Figure 2, we illustrate the (a,b) Kernel Smoothing density estimation of age and AOU measured in 1990s, 2000s, and 2010s, (c,d) Kernel Smoothing density estimation of Δ age and Δ AOU in the North Atlantic, (e, f) zonal mean of Δ age and AOU in the North Atlantic, comparing the 2000s to the 1990s, and (g, h) the 2010s to the 2000s. We also provided the quantitative analysis of Δ age and Δ AOU in the Table 1.

Revised Fig. 2:(a, b) Kernel Smoothing density estimates of water age (years) and Apparent Oxygen Utilization (AOU, $\mu\text{mol/kg}$) measured in the 1990s, 2000s, and 2010s. (c,d) Kernel Smoothing density estimation of the changes in water age (Δage) and AOU (ΔAOU) in the North Atlantic (blue: 2000s minus 1990s, red: 2010s minus 2000s). Zonal mean of Δage and AOU in the North Atlantic, comparing the 2000s to the 1990s (e, f), and the 2010s to the 2000s (g, h).

Tab.1: North Atlantic water age and Apparent Oxygen Utilization (AOU) and their temporal change across different periods. All measurements (see Fig. 1) are used in the analysis.

Variable (unit)	Periods	mean	median	mode	Volume weighted mean
Water age (yr)	1990	34.2	25.8	21.8	46.6
	2000	45.6	39.3	35.7	53.4
	2010	52.7	45.8	39.0	63.4
AOU (umol/kg)	1990	50.1	43.4	43.5	60.8
	2000	73.0	50.9	45.2	67.3
	2010	65.0	49.7	47.5	68.2
Δ Water age ^a (yr)	2000-1990	+3.9	+4.7	+5.8	+5.5
	2010-2000	+4.6	+6.0	+6.3	+6.6
Δ AOU ^a (umol/kg)	2000-1990	+1.8	+0.9	+0.5	+0.8
	2010-2000	+0.3	+0.7	+1.1	+1.1

^a Only in regions where the age and AOU are measured in both periods.

(2). The attribution of ventilation anomalies to climate change is not fully supported by the models. For one thing, not all models show consistent water age changes. For example, while some models show a water age decrease in the intermediate layer of the Labrador Sea, both EC-Earth3-CC and FOCI-MOPS show a water age increase (Figure S4). For another, even if the six models show a consistent water age change, the number of models (or ensemble members) is too small to completely exclude internal variability.

A: Thank you for highlighting this issue. We acknowledge that using only seven models may not fully capture internal variability. However, these are, to our knowledge, the only models available that uniquely simulate all necessary tracers for our analysis, including potential temperature, salinity, CFC-12, SF6, and dissolved oxygen concentration, and seven ensemble members might provide a useful impression of internal variability (even if not exhaustive). We look forward to more simulations of those tracers in the future. In the revised manuscript, we have conducted further analyses of water age and Apparent Oxygen Utilization (AOU) changes using Kernel density estimation across individual models and the multi-model mean (Fig. 3). Our results indicate that from the 1990s to the 2000s, North Atlantic waters generally became older and more oxygen-depleted in all models. Interestingly, the multi-model mean also reveals a decrease in water age at depths of 500m to 1300m from 40°N to 50°N during this period (Fig. 3c, d).

The supplementary materials provide comprehensive details on age and AOU anomalies for each model, confirming what the reviewer stated here, i.e. that not all models exhibit a decrease in water age at intermediate depths. In fact, we found the negative age and AOU anomalies, indicating younger waters, appear in different depths in different models (Fig. S3). Different states of internal variability in the different model solutions could be one reason for the

inter-model differences. We have emphasized in the revised manuscript that "Caution is advised when interpreting these patterns, as individual models display anomalies of younger water age at differing depths (Fig. S3). A distribution of counts for young and older anomalies across depths and isopycnals further confirms that individual models show different patterns in age differences (Fig. S4). Nonetheless, our analysis suggests that, despite younger age anomalies in parts of the North Atlantic, a larger volume exhibits signs of aging from the 1990s to the 2000s across most depths and isopycnals for all individual models, and consistent with observations. (Line 159-165)"

Revised Fig. 3: Panels (a and b) display the Kernel density estimation for Δ age and Δ AOU in the North Atlantic, comparing data from the 1990s to the 2000s across seven individual models and the model average. Panels (c and d) present the multi-model mean of the zonal average for Δ age and Δ AOU in the North Atlantic. The area covered by the multi-model mean (panels c and d) is smaller than that shown by the observational estimates (Fig. 2e,f), due to the slower spreading of abiotic transient tracers in models in the deep North Atlantic.

Tab.2: Temporal change (2000s minus 1990s) of North Atlantic water age and Apparent Oxygen Utilization (AOU) across multi-model mean and seven individual Earth System models.

Variable (unit)	Models	mean	median	mode	Volume weighted mean
Δ Water age (yr)	Multi-model mean	+2.3 \pm 1.2	+2.3 \pm 0.7	+1.7 \pm 1.1	+2.3 \pm 0.9
	CanESM5	+3.6	+2.6	+1.7	+3.8
	EC-Earth3-CC	+4.0	+3.4	+2.6	+2.6
	MRI-ESM2-0	+2.6	+2.5	+1.4	+2.2
	NorESM2-LM	+2.1	+2.5	+2.6	+2.7
	NorESM2-MM	+1.3	+2.6	+2.8	+2.4
	UKESM1-0-LL	+0.7	+1.2	-0.5	+1.1
	FOCI-MOPS	+1.7	+1.6	+1.6	+1.3
Δ AOU (mmol/kg)	Multi-model mean	+0.8 \pm 0.3	+0.7 \pm 0.4	+0.5 \pm 0.3	+0.6 \pm 0.2
	CanESM5	+0.5	+0.5	+0.6	+0.5
	EC-Earth3-CC	+1.3	+1.3	+0.8	+0.4
	MRI-ESM2-0	+0.4	+0.3	+0.2	+0.7
	NorESM2-LM	+0.8	+0.8	+0.9	+0.8
	NorESM2-MM	+0.9	+0.5	+0.4	+0.6
	UKESM1-0-LL	+1.1	+1.1	+0.2	+1.0
	FOCI-MOPS	+0.5	+0.2	+0.3	+0.4

Revised Fig. S3: Left and right panels present the zonal average for Δage (in unit of year) and ΔAOU (in unit of $\mu\text{mol/kg}$) in the North Atlantic from 1990s to 2000s.

Revised Fig.S4: Number of boxes indicating older (red) and younger (blue) water age and their difference (black = red minus blue) from the 1990s to the 2000s, plotted across (a) depths and (b) isopycnals. The solid lines depict the GLODAPv2.2022 data, while the dashed lines represent the multi-model mean. The dotted lines illustrate individual model results. Most of the volume in the North Atlantic exhibits signs of aging from the 1990s to the 2000s across most depths and isopycnals for all individual models, and consistent with observations

(3) The mechanism for ventilation changes under ssp 585 needs further elaboration. I am very confused on the enhanced ventilation of the intermediate waters under ssp585. If convective mixing weakens under global warming, both the intermediate and deep waters are expected to be less ventilated. Actually, when looking at Figure 4, I find water age increase for both intermediate and deep waters in the North Atlantic (50N-80N) for most of the models, suggesting weakened ventilation.

A: We have refined our analysis of the projected ventilation changes under the high CO₂ scenario and updated Fig. 4 accordingly. Our revised results confirm that ventilation is projected to weaken under warming conditions throughout the water column in the high-latitude North Atlantic (north of 50°N), as indicated by increases in both ideal water age and AOU across all models used (Fig. 4). We have corrected this in the manuscript (Lines 219-225).

However, despite this overall trend of weakening ventilation for the high-latitude North Atlantic, all models suggest increased ventilation at depths shallower than 500m in low-latitude regions (approximately south of 15N), as shown in Fig. 4 and Fig. S7. This phenomenon has been previously reported by Gnanadesikan et al. (2007) and Bopp et al., (2017), who proposed that

enhanced ventilation in low-latitude regions under warming scenarios results from reduced mixing with upwelled old waters. Additionally, Oschlies et al. (2018) offered an alternative explanation: the thinning of isopycnic layers in the upper ocean during a warmer climate may facilitate increased ventilation “speed” in mode waters (see their Fig. 2 for details). We discussed the above in lines 211-219 in the revised manuscript.

Revised Fig. 4: Panels (a) and (b) display the Kernel density estimates of Δage and ΔAOU in the North Atlantic, comparing data from the 2010s to the 2090s across individual models and the multi-model mean. Panels (c) and (d) show the multi-model mean of the zonal average for Δage and ΔAOU in the same region. In these analyses, the ideal age is used to detect ventilation changes; it functions like a "clock," resetting to zero at the sea surface and increasing by one day per day in the sub-surface layers. Note that, ideal age data in MRI-ESM-2-0 and dissolved oxygen concentration data in NorESM2-MM were not accessible for the SSP5-8.5 scenario during the course of our study (accessed in August of 2025, <https://esgf-metagrid.cloud.dkrz.de/search/cmip6-dkrz/>).

Revised Fig. S7: Left and right panels present the zonal average for Δage (in units of years) and ΔAOU (in units of $\mu\text{mol}/\text{kg}$) in the North Atlantic from the 2010s to the 2090s across Earth System models. The simulations are under high carbon dioxide emission scenario (SSP5-8.5). Note that, ideal age data in MRI-ESM-2-0 and dissolved oxygen concentration data in NorESM2-MM were not accessible for the SSP-585 scenario during the course of our study (accessed in August of 2025, <https://esgf-metagrid.cloud.dkrz.de/search/cmip6-dkrz/>).

Minor comments

(1). Lines 40-46: Ventilation does not equal to the AMOC. Most of the references mentioned here are measuring the AMOC, not ventilation.

A: We thank the reviewer for highlighting this important distinction. The reviewer is absolutely correct in noting that ventilation and the AMOC are not the same. Ventilation involves the exchange of properties of surface waters with the deeper ocean, playing a critical role in the cycling of oxygen and carbon. AMOC, however, is a large-scale circulation pattern focusing on the transport of water in the North Atlantic. Still, the ventilation of the Atlantic Ocean and the AMOC are interrelated. Because the AMOC includes the sinking of surface waters and its subsequent southward transport in the ocean interior, changes in AMOC are generally expected to influence ventilation.

We noted that the convection *depth* at high latitudes north of 50N is also very important for studying ventilation, and we have added more related references about it in the revised manuscript, including Lazier et al. (2002), van Aken et al. (2011), Rhein et al. (2017), and Yashayaev, I. (2024).

(2). Line 100 & Figure 2: The different water masses described in the text are indistinguishable from Figure 2. Please adjust the color maps for better illustration.

A: We use the section of "Water properties within the North Atlantic basin" to provide the background information about the water masses. However, as also suggested by other reviewers, we found that this part provided limited information on the temporal change of ocean ventilation in the North Atlantic that we focus on. Thus, in the revised manuscript, we removed this part and also this Figure.

(3). Lines 200-203: I do not think this conclusion can be made based on the very sparse observations globally. For example, in the subtropical North Atlantic, water age in the eastern basin indeed decreases. However, in the western subtropics, where the Subtropical Mode Water is formed, its age increases. That is to say, the spatial variation of water age is significant. With the sparse observations, conclusions need to be made with caution.

A: We agree that the age change in mode waters exhibits significant spatial variability. As the reviewer pointed out, water age increases in the eastern basin of the North Atlantic and decreases in the western basin of the North Atlantic from the 1990s to the 2000s. We have clarified this in the revised manuscript by stating, "While previous studies have also reported enhanced ventilation in certain mode or intermediate waters, such as Subantarctic Mode Water (Vaughn et al., 2013) and Arctic Intermediate Water (Jeasson et al., 2023), our findings highlight notable regional differences—for example, waters became younger in the upper eastern basin and older water in the upper western basin of the North Atlantic. More measurements of abiotic transient tracers over larger spatial and temporal scales are required for a better understanding of the change of ventilation in mode waters in the North Atlantic. "

(4). Figure 4: Which depths are the spatial distributions at in the right column?

A: In the original version, the right column panels showed the ideal age average over the entire depth domain of 0m-6000m. We have modified Fig. 4 in the revised manuscript to better illustrate the projected ventilation change in the North Atlantic.

(5). Line 224: I do not see a consistent water age change among all models. See my major comment (2).

A: We have refine our sentence in the revised manuscript.

(6). Method: The saturation of CFCs is assumed to be 100%. However, the saturation is likely variable in the mixed layer (see Figure 3 in Fine et al., 2017) and different between intermediate and deep waters. Sensitivity tests with variable saturation are suggested.

A: In our methodology, we initially assumed a 100% saturation of CFCs. However, recognizing the variability in saturation levels within the mixed layer and differences between intermediate and deep waters, we conducted sensitivity tests with variable saturation levels, as suggested. We explored saturation ranges from 50% to 100% in 10% increments. This covers the range of saturation levels reported in previous observational studies (e.g., Wallace and Lazier, 1988; Smethie and Fine, 2001; Raimondi et al., 2021).

Our findings indicate that applying a low saturation level, such as 50%, in the IG-TTD calculation renders water age calculations for depths shallower than 1500m in the Labrador Sea infeasible, as their partial pressure exceeds the highest atmospheric levels (refer to Fig. S2j). This indicates that assuming such low saturation levels for those waters is not plausible. Moreover, despite the high sensitivity of Δ age to saturation assumptions in Labrador Sea Water, the lower North Atlantic Deep Water consistently showed aging across all scenarios from the 1990s to the 2000s, and from the 2000s to the 2010s. We have added this analysis in our revised manuscript (Lines 131-139).

Revised Fig. S2: Δ age in the North Atlantic comparing data from the 1990s to the 2000s (a,c,e,g,i) and from the 2000s to the 2010s (b,d,f,h,j). Different assumptions about saturation levels are applied in the IG-TTD calculations of the individual panels.

Reference:

Bopp, L., Resplandy, L., Untersee, A., Le Mezo, P., & Kageyama, M. (2017). Ocean (de) oxygenation from the Last Glacial Maximum to the twenty-first century: insights from Earth System models. *Philosophical Transactions of the Royal Society A: Mathematical, Physical and Engineering Sciences*, 375(2102), 20160323. <https://doi.org/10.1098/rsta.2016.0323>

Gnanadesikan, A., & Russell, J. L. (2007). How does ocean ventilation change under global warming?. *Ocean Science*, 3(1), 43-53. <https://doi.org/10.5194/os-3-43-2007>

Lazier, J., Hendry, R., Clarke, A., Yashayaev, I., & Rhines, P. (2002). Convection and restratification in the Labrador Sea, 1990–2000. *Deep Sea Research Part I: Oceanographic Research Papers*, 49(10), 1819-1835. [https://doi.org/10.1016/S0967-0637\(02\)00064-X](https://doi.org/10.1016/S0967-0637(02)00064-X)

Oschlies, A., Brandt, P., Stramma, L., & Schmidtko, S. (2018). Drivers and mechanisms of ocean deoxygenation. *Nature geoscience*, 11(7), 467-473. <https://doi.org/10.1038/s41561-018-0152-2>

Raimondi, L., Tanhua, T., Azetsu-Scott, K., Yashayaev, I., & Wallace, D. W. (2021). A 30-year time series of transient tracer-based estimates of anthropogenic carbon in the Central Labrador Sea. *Journal of Geophysical Research: Oceans*, 126(5), e2020JC017092. <https://doi.org/10.1029/2020JC017092>

Rhein, M., Steinfeldt, R., Kieke, D., Stendardo, I., & Yashayaev, I. (2017). Ventilation variability of Labrador Sea Water and its impact on oxygen and anthropogenic carbon: a review. *Philosophical Transactions of the Royal Society A: Mathematical, Physical and Engineering Sciences*, 375(2102), 20160321. <https://doi.org/10.1098/rsta.2016.0321>

Smethie Jr, W. M., & Fine, R. A. (2001). Rates of North Atlantic Deep Water formation calculated from chlorofluorocarbon inventories. *Deep Sea Research Part I: Oceanographic Research Papers*, 48(1), 189-215. [https://doi.org/10.1016/S0967-0637\(00\)00048-0](https://doi.org/10.1016/S0967-0637(00)00048-0)

van Aken, H. M., de Jong, M. F., & Yashayaev, I. (2011). Decadal and multi-decadal variability of Labrador Sea Water in the north-western North Atlantic Ocean derived from tracer distributions: Heat budget, ventilation, and advection. *Deep Sea Research Part I: Oceanographic Research Papers*, 58(5), 505-523. <https://doi.org/10.1016/j.dsr.2011.02.008>

Wallace, D. W., & Lazier, J. R. (1988). Anthropogenic chlorofluoromethanes in newly formed Labrador Sea Water. *Nature*, 332(6159), 61-63. <https://doi.org/10.1038/332061a0>

Yashayaev, I., & Clarke, A. (2008). Evolution of North Atlantic water masses inferred from Labrador Sea salinity series. *Oceanography*, 21(1), 30-45. <https://www.jstor.org/stable/24860152>

Yashayaev, I. (2024). Intensification and shutdown of deep convection in the Labrador Sea were caused by changes in atmospheric and freshwater dynamics. *Communications Earth & Environment*, 5(1), 156. <https://doi.org/10.1038/s43247-024-01296-9>

Reference

Yashayaev, I. Intensification and shutdown of deep convection in the Labrador Sea were caused by changes in atmospheric and freshwater dynamics. *Commun Earth Environ* 5, 156 (2024). <https://doi.org/10.1038/s43247-024-01296-9>

Fine, R. A., S. Peacock, M. E. Maltrud, and F. O. Bryan (2017), A new look at ocean ventilation time scales and their uncertainties, *J. Geophys. Res. Oceans*, 122, 3771– 3798, doi:10.1002/2016JC012529.

Reviewer #2 Comments to Author:

Guo et al. present an analysis of historical transient tracer (CFC-12 and SF6) measurements from the North Atlantic to constrain an inverse model (Inverse Gaussian Transit Time Distribution; IG-TTD) to explore changes in apparent ventilation age since the early 1980s. Grouping the dataset (GLODAPv2) into three time periods, they find an overall increase in water age in the deep North Atlantic, and decrease in mean water age in the intermediate North Atlantic, suggesting a reduction in deep-ocean ventilation in the North Atlantic. The sign of deep- and intermediate-ocean changes in ventilation ages are consistent with simulations from several Earth System models, but in terms of ideal mean age (simulated directly from the models) and the age inferred from applying the IG-TTD approach to model-simulated CFC-12 and SF6. Overall the paper is well written and presented in a rather accessible format. The conclusions appear to be robust, but I would prefer to see (a) a more quantitative analysis of changes in inferred water mass age in the deep and intermediate North Atlantic, and (b) a more complete error analysis. I realize that much of the IG-TTD method and associated analysis is presented in a companion paper (Guo et al., 2024; preprint available via EGU sphere), which is focused on large-scale changes in water mass age throughout the global ocean and includes an analysis of, e.g., the sensitivity of results to the assumption of complete saturation at the time of air-sea exchange, or the choice of Δ/Γ . However, for this paper and its specific focus changes in ventilation age in the climatically important North Atlantic, I would like to see more sensitivity testing to fully convey the uncertainty on these main findings. The manuscript is timely, broadly relevant to global climate change, and well formulated. I would recommend publication in Nature Communications after the authors address these issues, which I describe in more detail below.

1. The main conclusions about changes in intermediate and deep North Atlantic ventilation age and other water mass properties (e.g., potential density, salinity, AOU) are described qualitatively and shown (quantitatively) in the figures (e.g., Figs. 2 and 3). I think it would strengthen the overall conclusions of the paper to provide average values (and uncertainties) of these properties and ages – and their temporal changes – for the water masses being discussed. For example, by defining intermediate and deep waters using either depth or potential density, the authors could estimate a mean change in ventilation age for age water mass, along with corresponding mean changes in salinity, potential density, potential temperature, and AOU. It would be helpful and provide more confidence in the ultimate result for the authors to be able to state that, e.g., the mean ventilation age of the deep North Atlantic increased by $x \pm y$ years from the 1990s to 2010s. Similarly, providing mean changes in these properties in the ESMs would be very helpful, while recognizing that the timescale of ESM simulations is different.

A: We thank the reviewer for their thoughtful and constructive feedback on our work. In the revised manuscript, we have updated figures (Fig. 1-4) and tables (Tab. 1-2) to quantitatively illustrate the mean state and change of water age and AOU observed in the North Atlantic over the past decades. We found that from 1990s to 2000s and from 2000s to 2010s, the water in the North Atlantic generally became older by +5.5 yr and +6.6 yr (volume weighted mean), and more oxygen depleted with an AOU increase of +0.8 $\mu\text{mol}/\text{kg}$ and +1.1 $\mu\text{mol}/\text{kg}$. Other statistical metrics similarly reflect trends of aging and oxygen depletion, as detailed in Tab. 1. This is now included in lines 81 to 97.

We have performed a similar analysis on the ESM simulations over the period from the 1990s to the 2000s. In general, the modeled water also becomes older and more oxygen-depleted, though

the magnitude of these changes in water age is generally underestimated compared to the observations. For further details, please refer to our response to comment 3 below.

Revised Fig. 2:(a, b) Kernel Smoothing density estimates of water age (years) and Apparent Oxygen Utilization (AOU, umol/kg) measured in the 1990s, 2000s, and 2010s. (c,d) Kernel smoothing density estimation of the changes in water age (Δ age) and AOU (Δ AOU) in the North Atlantic (blue: 2000s minus 1990s, red: 2010s minus 2000s). Zonal mean of Δ age and AOU in the North Atlantic, comparing the 2000s to the 1990s (e, f), and the 2010s to the 2000s (g, h).

Revised Tab.1: North Atlantic water age and Apparent Oxygen Utilization (AOU) and their temporal change across different periods.

Variable	Periods	mean	median	mode	Volume weighted mean
Water age (yr)	1990	34.2	25.8	21.8	46.6
	2000	45.6	39.3	35.7	53.4
	2010	52.7	45.8	39.0	63.4
AOU (umol/kg)	1990	50.1	43.4	43.5	60.8
	2000	73.0	50.9	45.2	67.3
	2010	65.0	49.7	47.5	68.2
Δ Water age ^a (yr)	2000-1990	+3.9	+4.7	+5.8	+5.5
	2010-2000	+4.6	+6.0	+6.3	+6.6
Δ AOU ^a (umol/kg)	2000-1990	+1.8	+0.9	+0.5	+0.8
	2010-2000	+0.3	+0.7	+1.1	+1.1

^a Only in regions where the age and AOU are measured in both periods.

2. Relatedly, I would like to see a formal error analysis on the main result, accounting for the sensitivity of IG-TTD ages to key factors like the choice of initial saturation and Δ/Γ . Could a monte carlo simulation be performed to estimate the uncertainty in the changes of ventilation age (and pot. temp., pot. density, salinity, AOU) for mean water masses (e.g., deep and intermediate North Atlantic)? For example, carrying out many independent monte carlo simulations using a range of assumptions about Δ/Γ and saturation, while also adding random perturbations to CFC-12 and SF6 measurements to account for measurement uncertainty. This would be an important addition to evaluate the robustness of the main result, which is presently hard to do based solely visual representation of the results in the figures.

A: Many thanks for the suggestions. We have now performed 100 independent Monte Carlo simulations by adding random perturbations to CFC-12 measurements with a range of -3% to 3%, which is within the precision of the measurements (Bullister and Wisegarver, 2008). Our analysis (Fig. S1) shows that considering measurement uncertainties, no volume-weighted mean

of Δ water age in the North Atlantic is 3.9 ± 0.03 yr (mean \pm standard deviation) from 1990s to 2000s, and 4.6 ± 0.05 yr from 2000s to 2010s.

We also added the uncertainties from the saturation assumptions by applying saturation from 90% to 50% with 10% increments during the IG-TTD calculation (see also response to R1 comment 6), a range suggested by previous studies (e.g., Wallace and Lazier, 1988; Smethie and Fine, 2001; Raimondi et al., 2021). We found that applying a low saturation level, such as 50%, in the IG-TTD calculation renders water age calculations for depths shallower than 1500m in the Labrador Sea infeasible, as their partial pressure exceeds the highest atmospheric levels (refer to Fig. S2j). This indicates that assuming such low saturation levels for these waters is not plausible. Moreover, despite the high sensitivity of Δ age to saturation assumptions in Labrador Sea Water, the lower North Atlantic Deep Water consistently showed aging across all scenarios from 1990s to 2000s, and from 2000s to 2010s.

Revised Fig. S1: Kernel density estimates of Δ age in the North Atlantic: from the 1990s to the 2000s (blue) and from the 2000s to the 2010s (red). Random measurement uncertainties within 3% have been incorporated. The plots are based on 100 Monte Carlo simulations.

Revised Fig. S2: Δ age in the North Atlantic comparing data from the 1990s to the 2000s (a,c,e,g,i) and from the 2000s to the 2010s (b,d,f,h,j). Different assumptions about saturation levels are applied in the IG-TTD calculations of the individual panels.

3. Are ESM simulations available running from the pre-industrial era through 2100? Or just 2015 to 2100? If the former are available, this would be a more useful comparison with the observational data set (e.g., binning ESM results into the same three time periods from the 1980s to present)

A: The ESM simulations are available from the pre-industrial era through 2100. We provided the Kernel density estimation of Δ age and Δ AOU in the North Atlantic in individual models and multi-model mean from 1990s to 2000s (Fig. 3) and also other statistical metrics of Δ water age

and ΔAOU (Tab. 2). We note that since the ESM only simulates CFC-12/SF6 until the year 2014, we are only able to provide the water age and AOU anomaly between the 1990s and 2000s.

Our analysis of future projections aims to further investigate how the ocean ventilation responds to climate change. Our results (see Fig. 4 in the revised manuscript) confirm that with amplified climate change, ocean ventilation in the North Atlantic -especially in the deep and high-latitude (north of 50N) regions - generally slows down. We also found that ventilation becomes stronger in tropical upper waters, as reported by previous studies (e.g., Gnanadesikan et al. 2007 and Bopp et al., 2017). Such enhanced ventilation in low latitudes is attributed to reduced mixing with upwelled old waters and also the thinning of shallow isopynic layers (Oschlies et al., 2018).

Revised Fig. 3 Panels (a,b) illustrate Kernel density estimation of Δage and ΔAOU in the North Atlantic in individual models and multi-model mean, (c, d) zonal mean of Δage and ΔAOU in the North Atlantic of the 2000s minus the 1990s. The area covered by the multi-model mean (panels c and d) is smaller than that shown by the observational estimates (Fig. 2e,f), due to the slower spreading of abiotic transient tracers in models in the deep North Atlantic.

Tab.2: Temporal change (2000s minus 1990s) of North Atlantic water age and Apparent Oxygen Utilization (AOU) across multi-model mean and seven individual Earth System models.

Variable	Moldes	mean	median	mode	Volume weighted mean
Δ Water age (yr)	Multi-model mean	+2.3 \pm 1.2	+2.3 \pm 0.7	+1.7 \pm 1.1	+2.3 \pm 0.9
	CanESM5	+3.6	+2.6	+1.7	+3.8
	EC-Earth3-CC	+4.0	+3.4	+2.6	+2.6
	MRI-ESM2-0	+2.6	+2.5	+1.4	+2.2
	NorESM2-LM	+2.1	+2.5	+2.6	+2.7
	NorESM2-MM	+1.3	+2.6	+2.8	+2.4
	UKESM1-0-LL	+0.7	+1.2	-0.5	+1.1
	FOCI-MOPS	+1.7	+1.6	+1.6	+1.3
Δ AOU (μ mol/kg)	Multi-model mean	+0.8 \pm 0.3	+0.7 \pm 0.4	+0.5 \pm 0.3	+0.6 \pm 0.2
	CanESM5	+0.5	+0.5	+0.6	+0.5
	EC-Earth3-CC	+1.3	+1.3	+0.8	+0.4
	MRI-ESM2-0	+0.4	+0.3	+0.2	+0.7
	NorESM2-LM	+0.8	+0.8	+0.9	+0.8
	NorESM2-MM	+0.9	+0.5	+0.4	+0.6
	UKESM1-0-LL	+1.1	+1.1	+0.2	+1.0
	FOCI-MOPS	+0.5	+0.2	+0.3	+0.4

Reviewer #3 Comments to Author:

This study examines changes in the ventilation system of the North Atlantic over the past few centuries by combining simulations and observations of transient tracers and dissolved oxygen. The authors adopt an interesting approach, using different tracers to represent water age, to address a pertinent research question regarding ventilation changes. However, I have major concerns about the manuscript in its current form that need to be addressed before it can be considered for publication.

The manuscript is overly descriptive, and the discussion and main conclusions are not adequately supported by the presented results. All trends and changes are discussed based on a few contour plots, which leave a lot of room for interpretation, especially since the observational plots are rather noisy (or variable).

A: Thank you for your constructive and thoughtful feedback. We acknowledge that the original manuscript was overly descriptive, with limited quantitative support for the main conclusions. In the revised version, we have addressed this by adding more robust, quantitative analyses. Additionally, in the original submission, our focus was spread across multiple hydrographic variables such as temperature and salinity, which may have diluted the emphasis on ventilation. In the updated manuscript, we have shifted our focus specifically to water age and AOU, providing clearer and more direct insights into ventilation changes. We believe these revisions strengthen the manuscript and better support our key findings.

For instance, the authors discuss trends in temperature, salinity, water age, and Apparent Oxygen Utilization (AOU) in different depth ranges and regions of the North Atlantic based on Figure 3a-j. They identify "significant" (lines 145, 150) and "slight" (line 151) changes in some regions, while in others, some properties have "become lower as expected" (line 142) or remained "relatively stable" (line 150). This description is not only vague but also not clearly visible in Figure 3. For example, the authors refer to "warmer and saltier anomalies in the upper Eastern Atlantic and colder, fresher anomalies in the upper Western Atlantic" (lines 136-137). While there is some blue (fresher, colder) in parts of the upper Western Atlantic and some red (warmer, saltier) in parts of the upper Eastern Atlantic, there is also a considerable amount of the opposite color present in these regions.

A: We thank the reviewer for your feedback regarding the description and reference to Figure 3. We have updated Figure 3 to a new Figure 2, and are focusing on the water age and AOU with more quantitative analysis (Tab. 1) in the revised manuscript. We found that from the 1990s to the 2000s and from the 2000s to the 2010s, the water in the North Atlantic generally became older by +5.5 yr and +6.6 yr (volume weighted mean), and more oxygen-depleted with AOU increased by +0.8 $\mu\text{mol/kg}$ and +1.1 $\mu\text{mol/kg}$ (Line 92 - 97).

Revised Fig. 2: (a, b) Kernel smoothing density estimates of water age (years) and Apparent Oxygen Utilization (AOU, $\mu\text{mol/kg}$) measured in the 1990s, 2000s, and 2010s. (c,d) Kernel smoothing density estimation of the changes in water age (Δage) and AOU (ΔAOU) in the North Atlantic (blue: 2000s minus 1990s, red: 2010s minus 2000s). Zonal mean of Δage and AOU in the North Atlantic, comparing the 2000s to the 1990s (e, f), and the 2010s to the 2000s (g, h).

Tab.1: North Atlantic water age and Apparent Oxygen Utilization (AOU) and their temporal change across different periods.

Variable	Periods	mean	median	mode	Volume weighted mean
Water age (yr)	1990	34.2	25.8	21.8	46.6
	2000	45.6	39.3	35.7	53.4
	2010	52.7	45.8	39.0	63.4
AOU (umol/kg)	1990	50.1	43.4	43.5	60.8
	2000	73.0	50.9	45.2	67.3
	2010	65.0	49.7	47.5	68.2
Δ Water age ^a (yr)	2000-1990	+3.9	+4.7	+5.8	+5.5
	2010-2000	+4.6	+6.0	+6.3	+6.6
Δ AOU ^a (umol/kg)	2000-1990	+1.8	+0.9	+0.5	+0.8
	2010-2000	+0.3	+0.7	+1.1	+1.1

^a Only in regions where the age and AOU are measured in both periods.

The same issue applies to changes in water age and AOU (lines 140-148). For example, the authors state that in the Labrador Sea, waters at 1000-2000 m depth have become younger, whereas waters at 1500-2500 m depth have become older. Again, while there is some blue (younger) and red (older) in these contour plots, it appears that waters between 500 and 1000 m have become younger, while the rest have become older, and even this is only true for a certain latitude band in the Labrador Sea (at least, based on the results shown in Figure 3).

A: We thank the reviewer for pointing out this error. We have clarified in the revised manuscript that, in the Labrador Sea, water age decreases occur at depths of 800m to 1200m during the 1990s to 2000s and from 1000m to approximately 1500m during the 2000s to 2010s. At lower latitudes, the presence of younger water is observed reaching depths of 2000m at several locations. Furthermore, we agree and our results show that the water age at depths of 1500m to 2500m in the Labrador Sea increased from the 1990s to the 2000s. We also categorized the count of grid boxes showing younger age or older age (Fig. S4). In conclusion, our qualitative interpretation remains robust as our analysis suggests that, despite younger age anomalies in parts of the North Atlantic (mainly in the upper Labrador Sea), most of the volume exhibits signs of aging from the 1990s to the 2000s across most depths and isopycnals, according to observations and all individual models.

Revised Fig.S4: Number of boxes indicating older (red) and younger (blue) water age and their difference (red minus blue) from the 1990s to the 2000s, plotted across (a) depths and (b) isopycnals. The solid lines depict the GLODAPv2.2022 data, while the dashed lines represent the multi-model mean. The dotted lines illustrate individual model results.

I observe similar problems and a lack of rigor in the discussion of the model results. For example, the authors claim that in all models, water age inferred from transient tracers and from the ideal age tracer are "consistent [...] except for minor differences in magnitude" (lines 166-168). However, the contour plots in Figure S4 show large regions where transient tracer and ideal age tracer suggest entirely different trends (e.g., for UKESM1-0-LL, the transient tracer suggests younger waters south of 40°N, whereas the ideal age tracer indicates older waters in the same region).

A: We suggest that the trend in ideal age observed in UKESM1-0-LL may be influenced by model drift. When examining the maximum ideal age in UKESM1-0-LL during the 2000s, we find it to be only around 160 years in the deep Pacific, which is unrealistically young for deep waters. This indicates that the ideal age may not have reached quasi-equilibrium due to insufficient spin-up time. Such a drift over a short period (ten years) is difficult to correct for.

In models, changes in AOU are typically primarily driven by variations in ventilation (e.g., Bopp et al., 2017; Buchanan and Tagliabue, 2021). Therefore, monitoring changes in AOU can provide insights into ventilation state changes. Our analysis shows that in 80% of sampled regions in the North Atlantic, the change of age and AOU shows the same sign. We also identified some noise and regional discrepancies (such as in parts of the intermediate oceans in MRI-ESM2-0 and the deep ocean in EC-Earth3-CC and FOCI), which we explain in the responses below.

Revised Fig. S3: Left and right panels present the zonal average for Δage (in unit of year) and ΔAOU (in unit of $\mu\text{mol}/\text{kg}$) in the North Atlantic from the 1990s to the 2000s. The area covered by models is smaller than that shown by the observational estimates (Fig. 2e,f), due to the slower spreading of abiotic transient tracers in models in the deep North Atlantic.

Observed North Atlantic Δ Age (year) versus Δ AOU ($\mu\text{mol}/\text{kg}$) from the 1990s to the 2000s (a) and from the 2000s to the 2010s (b). The numbers indicate the count of grid boxes displaying each combination of increased or decreased age and AOU: increased age with increased AOU, increased age with decreased AOU, decreased age with decreased AOU, and decreased age with increased AOU.

The same as the above figure (a), but for models.

Another example is the discussion on anthropogenic effects on ventilation (lines 189-194). Based on Figures 3 and S8, the authors describe largely persistent age anomaly patterns, with intermediate waters continuously becoming younger. However, there seem to be notable differences in these patterns that are omitted in the discussion. For example, the Labrador Sea Water (LSW) age anomaly at 1500-2500 m depth changes from positive (2000s minus 1990s; Fig. 3g) to negative (2010s minus 2000s; Fig. S8g), indicating no clear continuous trend. These are just a few examples. The overall presentation of the results and discussion is quite descriptive, and in parts, it might give the impression of being speculative. While I believe the authors have compiled a very valuable dataset that can provide important insights into ventilation patterns in the North Atlantic, the discussion and analysis of these results based solely on a few contour plots are insufficient. More in-depth analysis and quantitative plots would help the authors to better identify these trends and quantify them. For example, changes in temperature and salinity could be better illustrated using T-S plots, and changes in water age and AOU might be better shown in histograms for different periods and depth ranges, respectively.

A: We appreciate the reviewer's careful analysis and the suggestions. We point out the critical role of decadal variability in the ventilation process, particularly in regions like the Labrador Sea where intermittent deep convection events can obscure forced trends over shorter timescales. In the revised manuscript, we confirm that ventilation strength in the Labrador Sea exhibits large fluctuations from the 1990s to the 2000s, as pointed out by the reviewer and earlier studies (Yashayaev and Clarke, 2008; Yashayaev I, 2024). We have discussed it in the revised manuscript (lines 98-111). We have also conducted a more quantitative analysis on the Δ age and Δ AOU in the revised manuscript. Nonetheless, we would like to reemphasize that despite large fluctuations of ventilation in the Labrador Sea, most of the volume in the North Atlantic exhibits an aging trend.

Other (minor) comments:

- The panels in Figure 2 are too small, and the color ranges are too broad to distinguish the different water masses. For example, in the West Atlantic and Labrador Sea, LSW, uNADW, and INADW appear as indistinguishable blue (temperature) and green (salinity) blobs. Making the panels larger or adding specific isolines might help.

A: We intended to use this figure to provide the background information about the water masses, and in the revised manuscript, we removed this part as it provided limited information on the temporal change of ocean ventilation in the North Atlantic.

- Line 32: "the freshwater release" – Please specify which freshwater release or remove the article.

A: We have removed this part as suggested.

- Lines 71-99: This paragraph would be more appropriate in the Methods section rather than the Introduction.

A: We have shortened this paragraph into one sentence (lines 68 to 70) and moved the details into the Methods section as suggested.

- Lines 97-98: "Idealized age tracer simulations hence help to quantify the uncertainties of the IG-TTD approach." – This statement suggests that uncertainties of the method are quantified, but I have not seen this in the manuscript.

A: We have noticed that ideal age shares the significant drift problem in some models (e.g, UKSEM-LL). Thus, we provide a more quantitative analysis of Δ age, examining its sensitivity to measurement uncertainties and different saturation level assumptions (lines 128-146).

- Line 100: Section "Water properties within the North Atlantic basin" – It is unclear whether this section provides background information (based on literature) or presents results. If it presents results, Figure 2 needs revision. In its current form, the different water masses are indistinguishable due to the small panel size and broad colorbar ranges. T-S plots or histograms might better illustrate (changes in) water mass properties (see main comment above).

A: We intended to use this section to provide the background information about the water masses, and in the revised manuscript, we removed this part as it provided limited information on the temporal change of ocean ventilation in the North Atlantic.

- Line 109: Add a space between "28.0" and "kg".

A: We have removed this section.

- Lines 110-112: The phrase "Labrador Sea" is repeated three times in one short sentence.

A: We have removed this section.

- Line 140: Remove the space after "1000-".

A: We have removed this section for the reason described above.

- Figure 2: Please specify the reference pressure for potential density.

A: In the original manuscript, we used zero dbar as the reference pressure for potential density. In the revised manuscript, we have removed the original Figure 2.

- Figure 2: Consider marking regions with "old" waters, i.e., tracer concentrations below detection limit, and regions with no samples using different colors (currently both are white).

A: We have removed the original Figure 2.

- Lines 152-153: The differences between water age increase and AOU increase should be discussed or, at least, commented on further. While the detailed reasons for these differences might be complex and beyond the scope of this manuscript, both AOU and water age are crucial

tools in this study. The discrepancies are notable not only in deep waters (>2500 m) but also in large parts of the West Atlantic, where negative AOU anomalies (blue in Fig 3i) suggest better-ventilated/younger waters, contrary to what water age suggests (red in Fig 3g). It is problematic to say we trust these patterns only where they align with expectations and ignore them elsewhere.

A: We have observed that in our results, changes in water age and AOU sometimes exhibit different signs in certain parts of the ocean. While most observations indicate that increases in AOU are accompanied by increases in water age, approximately 35% of all the repeated grid boxes in the North Atlantic show opposing trends. In the models, this percentage is generally lower, with a mean of around 20% and individual models ranging from 18% to 34%. In fact, we found that previous studies also show the same phenomena. For example, Jeansson et al. (2023) compared the evolution of AOU and mean age in the Nordic Sea and demonstrated that some regions exhibit decreasing AOU alongside increasing mean age (see their Fig. 8).

In the revised manuscript (lines 113 to 126) we discuss more details on why does AOU sometimes increases while age decreases. We propose two possible explanations. First, changes in ocean circulation within a warming climate could alter water mass composition, as different water masses with varying biogeochemical histories are recombined over time—potentially introducing younger yet more oxygen-depleted waters into a given region. Second, local biological activity may influence the AOU signal independently of ventilation. For instance, even if ventilation slightly increases, an increase in local respiration rates could cause AOU to rise, reflecting biological consumption rather than changes in physical mixing. We also agree that discrepancies of age trend and AOU trend are meaningful and should be considered in the interpretation of ventilation patterns, rather than simply trusting only those that align with expectations.

- Lines 210-212: This argument seems oversimplified. To strengthen it, consider checking the NAO phase in the different models, possibly referencing existing literature.

A: We have added the North Atlantic Oscillation (NAO) index in individual models in Fig. S6. The NAO index is defined as the principal component of the dominant Empirical Orthogonal Function (EOF) pattern of winter (DJF: December-February) sea-level pressure over the North Atlantic region (20N–80N, 90W–40E). The NAO index are not always in the same phase across different models, indicating different internal variabilities of the used climate models.

Revised Fig. S6: The North Atlantic Oscillation (NAO) index in individual models. The NAO index is defined as the principal component of the dominant Empirical Orthogonal Function (EOF) pattern of winter (DJF: December-February) sea-level pressure over the North Atlantic region (20N–80N, 90W–40E). The NAO index are not always in the same phase across different models, indicating different internal variabilities of the used climate models.

• Figure 4: Why do the left panels have such different colorbar ranges (a factor of 10 difference between the top and bottom rows)?

A: The left panels in the original Fig. 4 have been removed to better focus on the temporal changes in ocean ventilation. We provide our answer regarding the ideal age difference between models, which is up to a factor of 10 between the CanESM5 (up to 2000 years) and UKESM1-0-LL (maximum 250 years). We propose that the insufficient spin-up of the ideal age in the UKESM1-0-LL is the primary reason for the unrealistically young deep waters. Ideal age is commonly initialized with zeros everywhere in the ocean and requires several thousand years to reach an equilibrium state (Wunsch and Heimbach, 2008). As the waters can only be as old as the length of the spin-up, thus, short model spin-up results in an unrealistic young ideal age.

Notably, we also identified that waters in CanESM5 are too old compared with the other models and also observations, indicated by age derived from CFC-12 and SF6 (figure below). Unlike the ideal age, the age derived from CFC-12 and SF6 is not influenced by the drift.

Age difference between CanESM5 with observations and other models. Age here refers to the mean age of IG-TTD derived from CFC-12 and SF₆

- Line 311: Is it possible to quantify, to some degree, the uncertainties of the IG-TTD age estimates? The plotted age anomalies are often small, and the spatial patterns are quite noisy/variable, raising questions about the robustness and significance of these changes.

A: In the revised manuscript, we estimate uncertainties in water age changes by considering both measurement and methodological uncertainties related to saturation assumptions. We refer details to the lines 121-139. In brief, we found that measurement uncertainties have a negligible effect on the detected temporal changes in water age. As for the saturation uncertainties, despite the high sensitivity of Δ age to saturation assumptions in Labrador Sea Water, the lower North Atlantic Deep Water consistently showed aging across all scenarios from the 1990s to the 2010s.

- Line 327: Should it say "ocean models" instead of "model oceans"?

A: We have changed into "ocean models".

- Line 329: Remove "are".

A: We have removed "are" as suggested.

- SI: Consider removing the date and time from the bottom of the pages.

A: We have removed the date and time from the bottom of the pages in SI.

The authors thank the reviewers for their very helpful comments and suggestions. We here provide a point-by-point response to all reviewers. The reviewers' comments are given in black, and our responses are given in blue. We refer to line numbers in our revised manuscript.

Reviewer #1 Comments to Author:

In the revised manuscript, authors have sharpened the focus on water age and AOU and included discussions on uncertainties regarding temporal sampling and saturation levels, which enhances the clarity of the manuscript and the robustness of the findings. Even so, the conclusions could be strengthened by addressing the following key points:

1. The discussions on water age decadal variability and trend are better explained in the revised manuscript. While the decadal contrast between 1990s and 2000s is quite clear, that between 2000s and 2010s remains less convincing. The top reason is that data used for 2010s is from 2015-2021, covering only 7 years. This time period is too short to robustly represent the mean ventilation strength during the decade, given that the transit time for deep ventilated waters to lower latitudes exceeds 10 years (e.g. an average export time scale for overflow waters is ~20 years based on high-resolution ocean model; Lozier et al., 2013). To address this inherent limitation of the observational record, the authors could use model output to reconstruct a longer historical time series of deep water age or AOU to better contextualize the trend.

A: We thank the reviewer for this insightful comment. We fully agree that the 2015–2021 observational period is too short to robustly capture decadal-scale variability, and that the deep convection initiated in 2015 have not yet propagated into the deep ocean. To address this limitation, we extended our analysis using age and AOU anomalies in models over longer periods until the year 2100 (lines 249-266). This modeling approach provides additional context for interpreting the observed patterns and allows us to assess the long-term evolution of deep water properties beyond the available observational record. Consistent with the observations, the model simulations indicate that the aging signal in the North Atlantic persists through the 21st century, reinforcing our conclusions regarding the overall trend.

2. While the uncertainty from temporal sampling is now discussed, the potentially significant impact of spatial sampling biases—raised in the previous review—warrants similar investigation. Testing the results against model data subsampled at the observational spatial locations would greatly strengthen the analysis.

A: We agree that comparing subsampled and full-field model outputs helps assess whether the signals detected at observational locations are representative of basin-scale ventilation changes. To address this, we compared the subsampled Δ age and full-field Δ age (60°W–15°W, 0–65°N) from the 1990s to the 2000s, as shown below. We found that the Δ age over the full field is still positive in all models, with a multi-model mean of $+2.1 \pm 1.1$ years. For individual models, although the full-field Δ age in UKESM1-0-LL differs noticeably from its subsampled counterpart at low latitudes, the vertical Δ age patterns are very similar in most models.

Importantly, we also emphasize that, due to the relatively short atmospheric history of CFC-12, this tracer cannot effectively constrain the age of old waters. This limitation, previously described in the *Methods* section, is now explicitly discussed in the main text (lines 84-91) :

“Despite the methodological advancements and the advantages of using CFC-12, it is important to note that, due to the relatively short atmospheric history of CFC-12, this tracer cannot effectively constrain the age of very old waters. We therefore focus on waters younger than 200 years, consistent with previous studies (Sulpis et al., 2021). To assess ventilation changes in older water masses, additional measurements and simulations involving other tracers, such as Argon-39 and radiocarbon, will be required. However, these tracers are currently either not measured with sufficient spatial and temporal coverage (Ebser et al., 2018) or not yet widely implemented in the CMIP models.”

Full-field (left panels; 60°W–15°W, 0–65°N) and subsampled (right panels) Δ age from the 1990s to the 2000s. Numbers in the left panels indicate the mean Δ age values.

3. In observations, aging trend is dominated by waters below 2000m, where model output is mostly lacking. Instead, aging trend in models is concentrated in the upper 1000m. This model-observation difference needs to be elaborated because it directly relates to the type of water mass experiencing ventilation change (e.g. mode water or deep/abyssal water). Quantifying and comparing trends above and below 2000m separately would help clarify the dominant layers of ventilation change and reconcile this difference.

A: Following your suggestion, we compared the age and AOU anomalies between the 1990s and 2000s in both models and observations, separating the ocean into two depth ranges: 0–1500 m and below 1500 m (Fig. S3). 1500m is chosen as it is the depth below which models clearly underestimate observed aging from visualization. In the upper 1500 m, the differences between modeled and observed Δage and ΔAOU are not statistically significant (simulated minus observed $\Delta\text{age} = -0.2$ yr; 95% CI [-2.4, 2.0]; $\Delta\text{AOU} = -0.7$ $\mu\text{mol kg}^{-1}$; 95% CI [-1.8, 0.3]). In contrast, the models underestimated the aging of deep waters below 1500 m, where simulated minus observed $\Delta\text{age} = -5.7$ yr (95% CI [-8.0, -3.3]) and $\Delta\text{AOU} = -1.1$ $\mu\text{mol kg}^{-1}$ (95% CI [-2.4, 0.1]). This discrepancy is primarily driven by the Labrador Sea region (80°W–30°W, 48°N–65°N), where the simulated Δage is lower than observed by about 9.8 yr (95% CI [-15.5, -4.1]) and ΔAOU is lower by -2.4 $\mu\text{mol kg}^{-1}$ (95% CI [-4.5, -0.3]) . We attribute this difference to the models' inability to reproduce the intense deep convection of the early 1990s and its subsequent weakening.

We have incorporated this analysis into the Discussion section on simulated ventilation changes in the Labrador Sea (lines 175 - 181):

“A detailed comparison between observed and simulated age anomalies suggests that the models fail to capture the pronounced age and AOU increases below 1500m at the Labrador Sea (48°N–60°N, 80°W–30°W) from the 1990s to the 2000s, likely due to their inability to simulate the intense deep convection of the early 1990s and its subsequent weakening. Further investigation is needed to confirm these discrepancies and understand their causes.”

Revised Fig. S3: Differences in Δage (left panels) and ΔAOU (right panels) between models and observations (model minus observation). Black lines separate the upper 1500 m from deeper layers. Numbers indicate the mean Δage and ΔAOU differences, with colors denoting whether models underestimate (blue) or overestimate (red) the aging signal.

Minor comments

4. Line 68: Specify Δ/Γ in text.

A: We have clarified the definition and specification of Δ/Γ in the revised manuscript (line 70). We refer to the details of how the Δ/Γ is constrained in the Method section.

5. Line 71: Specify what the modes are.

A: We have clarified it (lines 71 - 74).

6. Figure 2: Please clarify in figure caption that (a-b) are for all sections while (c-d) are for repeat sections.

A: We have clarified it in the revised manuscript.

7. Please acknowledge the limited number of models used may introduce uncertainties to the trend signal.

A: We agree that the limited number of models may introduce uncertainty in the trend estimates. This has been acknowledged in the revised discussion (lines 243 - 248):

“Notably, we used seven Earth System Models in this study, as to our knowledge these represent all available CMIP6 simulations that provide the necessary variables for our analysis. We acknowledge that the limited number of models may not fully capture internal variability and could introduce uncertainty in the estimated ventilation trends. Future studies employing larger model ensembles will help further constrain this uncertainty.”

Reference

Lozier, M. S., Gary, S. F., & Bower, A. S. (2013). Simulated pathways of the overflow waters in the North Atlantic: Subpolar to subtropical export. *Deep Sea Research Part II: Topical Studies in Oceanography*, 85, 147–153.

Reviewer #2 Comments to Author:

The authors have satisfactorily responded to my initial comments.

Reviewer #3 Comments to Author:

I appreciate the effort the authors have put into revising the manuscript and generally find my comments well addressed. The added, more quantitative analysis makes the manuscript much stronger.

That said, I share some of the concerns raised by Reviewer #1 in the first round regarding whether the presented results and analyzed data provide sufficient evidence to conclusively attribute the reported decadal ventilation changes in the North Atlantic to climate change. However, I deem the results interesting and worth publishing, as the authors have added critical discussion on the robustness of their results and the inherent limitations of available observations and model results, and have consequently phrased their conclusions more carefully. Considering this, I wonder whether it would be more appropriate to also reflect this in the title by phrasing it as a question, i.e., "Variation of ventilation in the North Atlantic over the past three decades - a climate change signal?"

A: We agree that, although the robustness of our findings has been evaluated to some extent, uncertainties remain in conclusively attributing the observed and simulated decadal ventilation changes in the North Atlantic to climate change. These uncertainties primarily arise from the relatively short observational period and the limited number of model simulations available for CFC-12, SF₆, and O₂ (only seven models). To better reflect these limitations and maintain a balanced interpretation, we have revised the title to include a question mark, as suggested.

Below are a few additional comments that I would appreciate seeing addressed:

Line 68: This seems to be the first usage of the symbols delta and gamma. Please explain here what they mean.

A: We introduce in line 70 that " Δ and Γ are the mean and width of the age spectrum, respectively (Vaugh et al., 2003)".

Line 70: "... we used all observational data into ..." >>> something is wrong with this sentence

A: Thank you for pointing this out. We have corrected the grammar to: "...we grouped all observational data into three periods..."

Lines 75-76: "Although we noted AOU as a measure of accumulated respiration that biological activities can influence (Buchanan & Tagliabue, 2021), it still contributes ..." >>> wording needs clarification

A: We have modified the sentence as " While AOU reflects accumulated respiration and can be influenced by biological activity (Buchanan & Tagliabue, 2021), it remains a valuable indicator for understanding changes in ventilation."

Line 85: Briefly explain or give reference to Kolmogorov-Smirnov test

A: We have added references for the two-sample Kolmogorov-Smirnov test: Massey, 1951; The MathWorks, Inc., 2024.

Figure 2a: Do you have an explanation why the changes from 1990s to 2000s are much more significant than from 2000s to 2010s? I presume it has to do with sampling locations? You mention this in lines 91-93, but I believe it could be made clearer.

A: Because here our analysis includes water age measurements from all available sections, the estimated probability distributions of water age are indeed influenced by variations in sampling locations and the associated ventilation across different decades. This effect causes the change in water age from the 1990s to the 2000s to appear larger. We have clarified it in the revised manuscript (lines 102 - 104):

“Notably, in addition to temporal changes in ocean ventilation, the age differences among decades are also influenced by variations in sampling locations (Fig. 1) and the associated ventilation states.”

Lines 109-111: Sentence “This underscores...” is somewhat unclear. I can guess the meaning, but the wording is rather confusing (e.g., what acts on shorter time scales here, and why should deep convection not be influenced by external forcing?)

A: Sorry for the confusion. Our intention was to emphasize that, similar to studies of AMOC decline, strong **natural** variability in deep convection can mask externally forced ventilation trends, particularly when assessed over observational periods shorter than or similar with the characteristic timescale of natural variability. To clarify this point, we have revised the sentence as follows:

“This underscores the critical role of natural decadal variability in the ventilation process of the Labrador Sea, where intermittent deep convection events can obscure trends driven by external forces over shorter timescales.”

Line 144: “such as 50%” instead of “like 50%” ?

A: We have changed “like 50%” to “such as 50%” as suggested.

Lines 155-156: The expression “In the mean of models ocean, ...” sounds wrong

A: We have corrected it as “In the multi-model mean...”

Lines 163-165: Could you provide some informed speculations about the potential reasons for these discrepancies in all models?

A: Reviewer 1 also asked the same question. In the revised manuscript, we compared the age and AOU anomalies between the 1990s and 2000s in both models and observations, separating the ocean into two depth ranges: 0–1500 m and below 1500 m (Fig. S3). In the upper 1500 m, the differences between modeled and observed Δ age and Δ AOU are not statistically significant

(simulated minus observed $\Delta\text{age} = -0.2 \text{ yr}$; 95% CI [-2.4, 2.0]; $\Delta\text{AOU} = -0.7 \mu\text{mol kg}^{-1}$; 95% CI [-1.8, 0.3]). In contrast, the models underestimated the aging of deep waters below 1500 m, where simulated minus observed $\Delta\text{age} = -5.7 \text{ yr}$ (95% CI [-8.0, -3.3]) and $\Delta\text{AOU} = -1.1 \mu\text{mol kg}^{-1}$ (95% CI [-2.4, 0.1]). This discrepancy is primarily driven by the Labrador Sea region (80°W–30°W, 48°N–65°N), where the simulated Δage is lower than observed by about 9.8 yr (95% CI [-15.5, -4.1]) and ΔAOU is lower by $-2.4 \mu\text{mol kg}^{-1}$ (95% CI [-4.5, -0.3]) . We attribute this difference to the models' inability to reproduce the intense deep convection of the early 1990s and its subsequent weakening.

We have incorporated this analysis into the Discussion section on simulated ventilation changes in the Labrador Sea (lines 175 - 181):

“A detailed comparison between observed and simulated age anomalies suggests that the models fail to capture the pronounced age and AOU increases below 1500m at the Labrador Sea (48°N-60°N, 80°W-30°W) from the 1990s to the 2000s, likely due to their inability to simulate the intense deep convection of the early 1990s and its subsequent weakening. Further investigation is needed to confirm these discrepancies and understand their causes.”

Revised Fig. S3: Differences in Δage (left panels) and ΔAOU (right panels) between models and observations (model minus observation). Black lines separate the upper 1500 m from deeper layers. Numbers indicate the mean Δage and ΔAOU differences, with colors denoting whether models underestimate (blue) or overestimate (red) the aging signal.

Lines 172-173: Concerning “The distribution of counts ...”: I am wondering if a vertically varying model resolution would in any way impact the shape of these distributions. Same for other distributions that are based on "counts", e.g., Fig. 2a-d. Or is this irrelevant because of how you bin the data to standard depths and how the distributions are then constructed?

A: The model resolution itself does not affect the shape of the distributions, as all model outputs were interpolated onto a $1^\circ \times 1^\circ$ horizontal grid and 33 standard depth levels. The regridded model fields were then subsampled according to the spatial and temporal locations of the observations, and the subsampled outputs were processed using the same binning procedure as the observational data. Consequently, the distributions based on counts were constructed using standardized bins, ensuring consistency between models and observations. However, if the vertical or horizontal resolution influences the representation of ventilation changes, such effects would indeed be reflected in the resulting distributions.

Figure 3a,b and similar KS-distributions: Consider adjusting the x axis range to make the relevant regions clearer as the pdfs are essentially 0 from ± 25 yr and ± 15 $\mu\text{mol/kg}$ outward. As it is, there is a lot of wasted white space in the figure and the relevant parts are unclear.

A: We have revised the figure as suggested.

Figure 3d: y axis label and tick labels overlap

A: We have adjusted the figure.

Caption of Figure 3: In second line, add space between “and” and “ Δ ”

A: We have added space.

Lines 186-187: “... temporal coverage of observations ...”

A: We have revised the sentence as suggested.

Lines 191-192: “sea ice loss” instead of “ice melting”? (as ice melting occurs naturally all the time)

A: Thanks for the thoughtful suggestion. We have revised the sentence as suggested.

Line 207 and other relevant passages throughout the manuscript: You generally refer to basin scale (as in full depth) mean water age. Is that really the best metric to assess ventilation changes? I would assume intermediate or deep water age might be more appropriate and might yield quite different values for the mean aging trends. Or is this irrelevant because of the way you calculate age? Could you please comment on this?

A: We thank the reviewer for this thoughtful comment. Our goal is to assess how ocean ventilation has changed in the North Atlantic and to interpret these changes in a basin-scale context. Therefore, we focused on the mean water age as an integrated metric to characterize large-scale ventilation variability and used model simulations—spanning longer timescales and exhibiting clearer anthropogenic signals from future projection—to aid interpretation. We acknowledge, however, that a depth-specific analysis will provide additional insight into layer-dependent ventilation processes (e.g., stronger ventilation in upper tropical waters), which will be explored in future work.

Line 204: Drop “For the real ocean,”

A: We have deleted “For the real ocean,”

Line 307: “... in the main text ...”

A: We have revised the sentence.

Lines 352-353: You discuss the limitations of the method in general, but do not elaborate what this implies for your results. For example, does this sentence imply that your age estimates should be seen as lower bounds and that the actual age is likely larger in regions where the real distributions are multi-modal?

A: In the previous version, this section was included to discuss the mean state of water age; however, we have removed it in the revised manuscript to avoid redundancy. As for the content included here, it is true that the mean age derived from the IG-TTD framework can be biased toward younger values due to imperfect assumptions about the shape of the transit time distribution and the relatively short atmospheric history of abiotic transient tracers (Guo et al., 2025, Ocean Science, DOI: 10.5194/os-21-1167-2025).

Regarding temporal changes that we focused on, rather than relying on single-tracer constrained IG-TTD mean ages which can introduce spurious trends, we applied a dual-tracer (CFC-12 and SF₆) constrained IG-TTD approach. This method provides a more reliable estimate of ventilation trends by better constraining the Δ/Γ ratio (Guo et al., 2025, Ocean Science).

Table 3: In the third row, fix citation and reference (line 431). “Consortium” is not a last name but part of the full name of the consortium. If I am not mistaken, this is the EC-Earth Consortium (EC-Earth)

A: Thanks for pointing out this error and we have corrected it.

The authors thank the reviewers for their very helpful comments and suggestions. We here provide a point-by-point response to all reviewers. The reviewers' comments are given in black, and our responses are given in blue. We refer to line numbers in our revised manuscript.

Reviewer #1 (Remarks to the Author):

Authors have nicely and carefully addressed my concerns. I thus recommend immediate publication.

Reviewer #3 (Remarks to the Author):

The authors have addressed all my previous comments. I do believe a question mark in the title would make it more accurate. However, as this is not supported by the journal, I will suggest the following alternatives but leave it up to the authors and the journal editors to work this out:

Variation of ventilation in the North Atlantic over the past three decades – ...
... Evidence for a climate change signal
... Indications of a climate change signal
... A potential climate change signal

A: We have updated our title to “**North Atlantic ventilation change over the past three decades is potentially driven by climate change**” to align with Nature Communications' requirements. This title concisely captures the main findings of the manuscript while maintaining appropriate caution given the limited temporal coverage of observations and the number of models used.

Below are a few additional minor comments/suggestions/corrections:

Main text:

Figure 1: Depth colorbar labels are barely readable

A: We have updated Figure 1 and increased the size of the color-bar labels to ensure they are clearly readable.

Figure 2a-d: Panel labeling overlaps with y-axis tick labels

A: We have updated Figure 2 to resolve the overlap between the panel labels and the y-axis tick labels.

Most figures where y-axis shows depth: depth >>> Depth [m]

A: We have updated the y-axis labels in all relevant figures accordingly.

Figure 4d: y-axis label overlaps with tick labels

A: We have updated Figure 2 to resolve the overlap between the panel labels

Minor aesthetic note: Panel labeling in figure captions is very inconsistent. E.g., Figure 2. (a, b); Figure 3. Panels (a and b); Figure 4. Panels (a) and (b)

A: We have updated all relevant figures accordingly.

SI:

Figure S3: Unclear over what periods the differences are calculated.

A: We have clarified the periods in the figure legend, specifying that the differences represent the 2000s minus the 1990s.

Figure S4 caption: typo “acorss”

A: We have corrected the typo.

Figure S6: “The same as Fig. 3” >>> Should this be Figure S4 with the revised figure numbering?

A: The reference is to Figure 3 in the main manuscript, not to a supplemental figure. Figure 3 presents the kernel-smoothed probability distributions of subsampled Δage and ΔAOU . Figure S6 shows the same distributions without subsampling, allowing us to assess whether the temporal coverage of the subsampled data in this well-sampled region is sufficient to detect robust long-term variability or trends.

Figure S7 caption: “The NAO index are” >>> mixed plural and singular

A: We have corrected this grammatical error.

Figure S8 caption: typo “acorss”

A: We have corrected the typo.

Review of “Variation of ventilation in the North Atlantic over the past three decades – a climate change signal” by Guo et al.

Summary

Using observed and simulated transient abiotic tracers, the study investigates changes of ventilation in the North Atlantic in the 1990s, 2000s and 2010s. They find enhanced ventilation of the intermediate waters and reduced ventilation of the deep waters since the 1990s. These decadal changes are further regarded as a climate change signal in response to external forcing.

The study focuses on an interesting topic, but I am not convinced that the decadal ventilation changes in the North Atlantic are a climate change signal. This is because: (1) observed hydrography clearly shows a reduction of ventilation of Labrador Sea intermediate waters in the 2000s, which is part of a significant decadal variability, contradicting the argument by the current study; and (2) climate model results are insufficient to separate internal variability from a climate change signal. As such, I cannot recommend publication. Below I list my major comments.

Major comments

(1). The reported enhanced ventilation in the intermediate layer of the Labrador Sea in the 2000s contrasts previous studies using hydrography. By comparing water age in the 2000s to the 1990s (Figure 3g), it is argued that age of intermediate water (1000-2000 m) decreases, suggesting an *enhanced* ventilation in the Labrador Sea. However, hydrography clearly shows a warmer, saltier and lighter intermediate layer in the 2000s, which reflects *weakened* ventilation. This reduced ventilation in the 2000s is well documented in many observational studies (e.g. Yashayaev, 2024).

Actually, the age change between 1000 m and 2000 m in the Labrador Sea shown in Figure 3g seems to be positive to me. The color scale makes it difficult to read.

(2). The attribution of ventilation anomalies to climate change is not fully supported by the models. For one thing, not all models show consistent water age changes. For example, while some models show a water age decrease in the intermediate layer of the Labrador Sea, both EC-Earth3-CC and FOCI-MOPS show a water age increase (Figure S4). For another, even if the six models show a consistent water age change, the number of models (or ensemble members) is too small to completely exclude internal variability.

(3) The mechanism for ventilation changes under ssp 585 needs further elaboration.

I am very confused on the enhanced ventilation of the intermediate waters under ssp585. If convective mixing weakens under global warming, both the intermediate and deep waters are expected to be less ventilated. Actually, when looking at Figure 4, I find water age increase for both intermediate and deep waters in the North Atlantic (50N-80N) for most of the models, suggesting weakened ventilation.

Minor comments

(1). Lines 40-46: Ventilation does not equal to the AMOC. Most of the references mentioned here are measuring the AMOC, not ventilation.

(2). Line 100 & Figure 2: The different water masses described in the text are indistinguishable from Figure 2. Please adjust the color maps for better illustration.

(3). Lines 200-203: I do not think this conclusion can be made based on the very sparse observations globally. For example, in the subtropical North Atlantic, water age in the eastern basin indeed decreases. However, in the western subtropics, where the Subtropical Mode Water is formed, its age increases. That is to say, the spatial variation of water age is significant. With the sparse observations, conclusions need to be made with caution.

(4). Figure 4: Which depths are the spatial distributions at in the right column?

(5). Line 224: I do not see a consistent water age change among all models. See my major comment (2).

(6). Method: The saturation of CFCs is assumed to be 100%. However, the saturation is likely variable in the mixed layer (see Figure 3 in Fine et al., 2017) and different between intermediate and deep waters. Sensitivity tests with variable saturation are suggested.

Reference

Yashayaev, I. Intensification and shutdown of deep convection in the Labrador Sea were caused by changes in atmospheric and freshwater dynamics. *Commun Earth Environ* **5**, 156 (2024). <https://doi.org/10.1038/s43247-024-01296-9>

Fine, R. A., S. Peacock, M. E. Maltrud, and F. O. Bryan (2017), A new look at ocean ventilation time scales and their uncertainties, *J. Geophys. Res. Oceans*, **122**, 3771–3798, doi:[10.1002/2016JC012529](https://doi.org/10.1002/2016JC012529).

This study examines changes in the ventilation system of the North Atlantic over the past few centuries by combining simulations and observations of transient tracers and dissolved oxygen. The authors adopt an interesting approach, using different tracers to represent water age, to address a pertinent research question regarding ventilation changes. However, I have major concerns about the manuscript in its current form that need to be addressed before it can be considered for publication.

The manuscript is overly descriptive, and the discussion and main conclusions are not adequately supported by the presented results. All trends and changes are discussed based on a few contour plots, which leave a lot of room for interpretation, especially since the observational plots are rather noisy (or variable).

For instance, the authors discuss trends in temperature, salinity, water age, and Apparent Oxygen Utilization (AOU) in different depth ranges and regions of the North Atlantic based on Figure 3a-j. They identify "significant" (lines 145, 150) and "slight" (line 151) changes in some regions, while in others, some properties have "become lower as expected" (line 142) or remained "relatively stable" (line 150). This description is not only vague but also not clearly visible in Figure 3. For example, the authors refer to "warmer and saltier anomalies in the upper Eastern Atlantic and colder, fresher anomalies in the upper Western Atlantic" (lines 136-137). While there is some blue (fresher, colder) in parts of the upper Western Atlantic and some red (warmer, saltier) in parts of the upper Eastern Atlantic, there is also a considerable amount of the opposite color present in these regions.

The same issue applies to changes in water age and AOU (lines 140-148). For example, the authors state that in the Labrador Sea, waters at 1000-2000 m depth have become younger, whereas waters at 1500-2500 m depth have become older. Again, while there is some blue (younger) and red (older) in these contour plots, it appears that waters between 500 and 1000 m have become younger, while the rest have become older, and even this is only true for a certain latitude band in the Labrador Sea (at least, based on the results shown in Figure 3).

I observe similar problems and a lack of rigor in the discussion of the model results. For example, the authors claim that in all models, water age inferred from transient tracers and from the ideal age tracer are "consistent [...] except for minor differences in magnitude" (lines 166-168). However, the contour plots in Figure S4 show large regions where transient tracer and ideal age tracer suggest entirely different trends (e.g., for UKESM1-0-LL, the transient tracer suggests younger waters south of 40°N, whereas the ideal age tracer indicates older waters in the same region).

Another example is the discussion on anthropogenic effects on ventilation (lines 189-194). Based on Figures 3 and S8, the authors describe largely persistent age anomaly patterns, with intermediate waters continuously becoming younger. However, there seem to be notable differences in these patterns that are omitted in the discussion. For example, the Labrador Sea Water (LSW) age anomaly at 1500-2500 m depth changes from positive (2000s minus 1990s; Fig. 3g) to negative (2010s minus 2000s; Fig. S8g), indicating no clear continuous trend.

These are just a few examples. The overall presentation of the results and discussion is quite descriptive, and in parts, it might give the impression of being speculative. While I believe the authors have compiled a very valuable dataset that can provide important insights into ventilation patterns in the North Atlantic, the discussion and analysis of these results based solely on a few contour plots are insufficient. More in-depth analysis and quantitative plots would help the authors to better identify these trends and quantify them. For example, changes in temperature and salinity could be better illustrated using T-S plots, and changes in water age and AOU might be better shown in histograms for different periods and depth ranges, respectively.

Other (minor) comments:

- The panels in Figure 2 are too small, and the color ranges are too broad to distinguish the different water masses. For example, in the West Atlantic and Labrador Sea, LSW, uNADW, and INADW appear as indistinguishable blue (temperature) and green (salinity) blobs. Making the panels larger or adding specific isolines might help.
- Line 32: "the freshwater release" – Please specify which freshwater release or remove the article.
- Lines 71-99: This paragraph would be more appropriate in the Methods section rather than the Introduction.
- Lines 97-98: "Idealized age tracer simulations hence help to quantify the uncertainties of the IG-TTD approach." – This statement suggests that uncertainties of the method are quantified, but I have not seen this in the manuscript.
- Line 100: Section "Water properties within the North Atlantic basin" – It is unclear whether this section provides background information (based on literature) or presents results. If it presents results, Figure 2 needs revision. In its current form, the different water masses are indistinguishable due to the small panel size and broad colorbar ranges. T-S plots or histograms might better illustrate (changes in) water mass properties (see main comment above).
- Line 109: Add a space between "28.0" and "kg".
- Lines 110-112: The phrase "Labrador Sea" is repeated three times in one short sentence.
- Line 140: Remove the space after "1000-".
- Figure 2: Please specify the reference pressure for potential density.
- Figure 2: Consider marking regions with "old" waters, i.e., tracer concentrations below detection limit, and regions with no samples using different colors (currently both are white).

- Lines 152-153: The differences between water age increase and AOU increase should be discussed or, at least, commented on further. While the detailed reasons for these differences might be complex and beyond the scope of this manuscript, both AOU and water age are crucial tools in this study. The discrepancies are notable not only in deep waters (>2500 m) but also in large parts of the West Atlantic, where negative AOU anomalies (blue in Fig 3i) suggest better-ventilated/younger waters, contrary to what water age suggests (red in Fig 3g). It is problematic to say we trust these patterns only where they align with expectations and ignore them elsewhere.
- Lines 210-212: This argument seems oversimplified. To strengthen it, consider checking the NAO phase in the different models, possibly referencing existing literature.
- Figure 4: Why do the left panels have such different colorbar ranges (a factor of 10 difference between the top and bottom rows)?
- Line 311: Is it possible to quantify, to some degree, the uncertainties of the IG-TTD age estimates? The plotted age anomalies are often small, and the spatial patterns are quite noisy/variable, raising questions about the robustness and significance of these changes.
- Line 327: Should it say "ocean models" instead of "model oceans"?
- Line 329: Remove "are".
- SI: Consider removing the date and time from the bottom of the pages.